# LUMA: Low-Dimension Unified Motion Alignment with Dual-Path Anchoring for Text-to-Motion Diffusion Model

## Abstract

While current diffusion-based models, typically built on U-Net architectures, have shown promising results on the text-to-motion generation task, they still suffer from semantic misalignment and kinematic artifacts. Through analysis, we identify severe gradient attenuation in the deep layers of the network as a key bottleneck, leading to insufficient learning of high-level features. To address this issue, we propose **LUMA** (*Low-dimension Unified Motion Alignment*), a text-to-motion diffusion model that incorporates dual-path anchoring to enhance semantic alignment. The first path incorporates a lightweight MoCLIP model trained via contrastive learning without relying on external data, offering semantic supervision in the temporal domain. The second path introduces complementary alignment signals in the frequency domain, extracted from low-frequency DCT components known for their rich semantic content. These two anchors are adaptively fused through a temporal modulation mechanism, allowing the model to progressively transition from coarse alignment to fine-grained semantic refinement throughout the denoising process. Experimental results on HumanML3D and KIT-ML demonstrate that LUMA achieves state-of-the-art performance, with FID scores of 0.035 and 0.123, respectively. Furthermore, LUMA accelerates convergence by $1.4\times$ compared to the baseline, making it an efficient and scalable solution for high-fidelity text-to-motion generation.

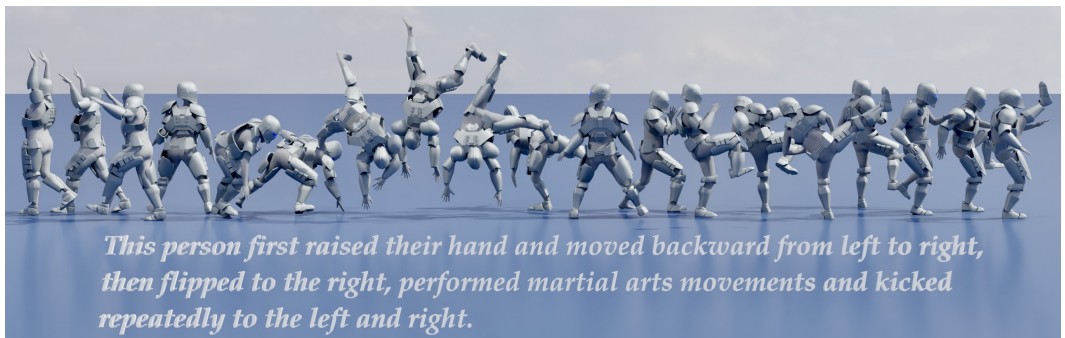

*This person first raised their hand and moved backward from left to right, then flipped to the right, performed martial arts movements and kicked repeatedly to the left and right.*

Figure 1: LUMA generates high-fidelity 3D human motion from natural language input. It injects temporal semantic anchor from the lightweight MoCLIP encoder and frequency semantic anchor from low-frequency DCT directly into the diffusion backbone. This design overcomes gradient attenuation in deep layers and achieves state-of-the-art motion fidelity.

## 1 Introduction

Text-driven human motion generation has recently garnered considerable attention due to the semantic richness and intuitive nature of natural language descriptions. This technology holds great potential for a wide range of applications, including animation, film production, virtual/augmented reality (VR/AR), and robotics Chen et al. (2024; 2025); Pinyoanuntapong et al. (2024c;a;b). Re-

cent performance gains have largely stemmed from the development of increasingly sophisticated architectures. Notable examples include ReMoDiffuse Zhang et al. (2023b), MotionLCM Dai et al. (2024) and MMM Pinyoanuntapong et al. (2024c). While these approaches have achieved notable success, they often rely on progressively larger and more complex models, resulting in diminishing efficiency and limited practical improvement.

Current research in text-to-motion generation mainly follows two paradigms. The first employs VAE-based models that encode motion into discrete tokens, predicted via autoregressive (AR) Zhang et al. (2023a); Jiang et al. (2023) or non-autoregressive (NAR) Guo et al. (2023); Pinyoanuntapong et al. (2024c;b) frameworks. The second leverages diffusion-based models, which iteratively transform Gaussian noise into motion sequences under text guidance Tevet et al. (2022); Chen et al. (2023); Huang et al. (2024); Zhang et al. (2023b); Dai et al. (2024). While VQ-based discrete methods have gained popularity Guo et al. (2023); Pinyoanuntapong et al. (2024b), they often suffer from tokenization-induced information loss and limited motion diversity Zhang et al. (2023a); Guo et al. (2022b). In contrast, diffusion models offer finer motion details, greater diversity, and improved physical plausibility, making them a promising alternative Yuan et al. (2023b); Zhang et al. (2022); Tevet et al. (2022); Chen et al. (2025). Notably, recent state-of-the-art diffusion models such as StableMotion Huang et al. (2024) and MLD Chen et al. (2023) adopt U-Net architectures as the core component for intermediate feature extraction and motion reconstruction. However, these models are limited by their high computational cost during training and the relatively low semantic fidelity of the generated motions.

Through experimental analysis, we identify the root cause of this issue as gradient shifts in the deeper layers of the U-Net. Specifically, we observe that the gradients in the downsampling and bottleneck layers are significantly smaller, and in some cases, nearly vanish compared to those in the upsampling layers. This phenomenon leads to the network's inability to effectively learn high-level abstract features from the input data Srivastava et al. (2015); He et al. (2016); Alzubaidi et al. (2021), resulting in slow or even stalled convergence during training. Consequently, the generated motion sequences often suffer from missing fine-grained details, incomplete structural representations, and inaccurate semantic alignment. These limitations ultimately hinder the overall performance of the model.

To address this issue, an intuitive approach is to introduce additional dense supervision signals at intermediate layers, rather than relying solely on sparse loss functions applied to the final output. Existing methods, such as Representational Alignment (REPA) Yu et al. (2025), significantly enhance semantic expressiveness by utilizing intermediate features from large-scale pre-trained models (e.g., DINOv2-g) Oquab et al. (2024) as additional supervisory signals. Nevertheless, REPA heavily relies on external pre-trained models, limiting its applicability in motion tasks lacking suitable pre-trained models or with limited high quality motion data.

While a wide range of options exists for selecting dense supervision signals Wang & He (2025), it is critical to ensure they effectively enhance the semantic representation capability of deep layers. To this end, we leverage supervision signals from both temporal and spatial–frequency dimensions. Consequently, we propose a dual-path unified semantic alignment framework that eliminates the reliance on large-scale pretrained model supervision. In the temporal path, we introduce *MoCLIP*, a lightweight contrastively trained text–motion encoder, to extract semantic representations directly from motion sequences without any external corpora or teachers. The second path introduces an orthogonal frequency domain alignment anchor which serves as a complementary counterpart to the temporal domain. Specifically, motion data is transformed into the frequency domain using Discrete Cosine Transform (DCT), and its low-frequency components, recognized for their rich semantic content, are used as stable and orthogonal supervisory signals. To effectively integrate these two complementary supervision signals, we design an adaptive FiLM Perez et al. (2017) modulation module that dynamically and progressively adjusts the strength of semantic feature injection throughout the network.

We summarize our contributions as follows:

    (i) We conduct the first systematic gradient analysis of diffusion-based text-to-motion models, revealing severe sparsity in down-sampling and bottleneck layers that hinders semantic learning and slows convergence.

(ii) We introduce *MoCLIP*, a new motion encoder for text–motion alignment. MoCLIP couples a transformer based motion encoder with a CLIP text encoder under a contrastive objective, and its token-wise text embeddings are fused into each U-Net block via cross-attention to strengthen semantic grounding without relying on large pretrained teachers.

(iii) We propose *Dual-Path Timestep-Aware Semantic Anchoring*, which injects complementary temporal and frequency semantic anchors via FiLM modulation to revitalize deep-layer gradients, eliminating the need for large-scale pretrained teachers.

(iv) Extensive experiments on multiple datasets demonstrate that our LUMA framework achieves state-of-the-art performance across various metrics, while converging significantly faster than baselines.

## 2 RELATED WORK

### 2.1 TEXT-TO-MOTION GENERATION.

Text-to-motion generation has been primarily driven by two dominant paradigms: diffusion-based denoising models and autoregressive token-based models. Diffusion frameworks such as MDM Tevet et al. (2022), MotionDiffuse Zhang et al. (2024a), and PhysDiff Yuan et al. (2023a) iteratively refine noise into motion using cross-attention and physics-guided denoisers, establishing strong baselines for realism. Latent diffusion variants, exemplified by MLD Chen et al. (2023) and MotionLCM Dai et al. (2024), compress motion trajectories into compact latent codes and achieve real-time sampling. Retrieval-based methods like ReMoDiffuse Zhang et al. (2023b) inject exemplar motions for style control, while control-oriented pipelines incorporate explicit constraints to improve diversity, alignment, and user steering.

On the other track, autoregressive approaches such as T2M-GPT Zhang et al. (2023a), MoMask Guo et al. (2023), BAMM Pinyoanuntapong et al. (2024b), InfiniMotion Zhang et al. (2024b), and TEACH Athanasiou et al. (2022) treat motion as token sequences and rely on transformer language models to generate coherent long-range structure. More recently, LaMP Li et al. (2025) revisits autoregressive modeling through language–motion pretraining, improving cross-modal alignment. However, these models are still trained using single-scale and low-level supervision, typically frame-wise MSE or token reconstruction. This weakens gradient propagation in deeper layers and leads to semantic dilution and loss of detail on complex prompts. Multi-scale and cross-modal objectives remain essential for addressing these limitations. Our work addresses this by introducing temporal and frequency semantic anchors, combined with timestep-aware modulation to guide the denoising process.

### 2.2 REPRESENTATION ALIGNMENT FOR DIFFUSION MODELS.

Recent studies have explored enhancing diffusion models by aligning their internal representations with high-level semantic features to improve learning efficiency and generation quality. Representation Alignment (REPA) Yu et al. (2025) explicitly aligns intermediate hidden states of diffusion transformers with semantically rich features from large-scale vision encoders (e.g., DINOv2, CLIP), yielding over 17× faster convergence and improved generation quality. Extensions rapidly broaden the scope: REPA-E Leng et al. (2025) unlocks end-to-end training by jointly tuning the VAE tokenizer alongside the diffusion backbone; U-REPA Tian et al. (2025) adapts the paradigm to U-Net architectures through spatial up-sampling and a manifold loss; and HASTE Wang et al. (2025) couples holistic (feature + attention) alignment with stage-wise termination to mitigate late-stage capacity mismatch. However, these methods rely on large-scale pretrained teachers and abundant paired vision data, which are unavailable for text-to-motion tasks. This makes it difficult to replicate the strong teacher–student setup that REPA relies on. Our work differs from existing representation alignment methods by removing the reliance on pretrained teachers and introducing temporal and frequency anchors that are learned jointly with the diffusion model through timestep-aware modulation. This design improves deep-layer gradients and maintains motion detail under limited supervision.

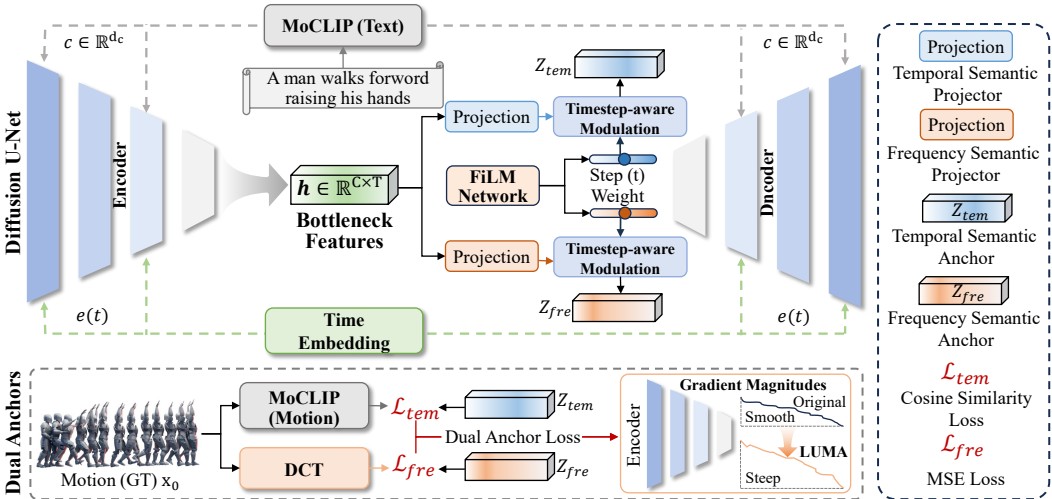

Figure 2: **Overview of the LUMA framework.** Text embeddings from MoCLIP are injected into each U-Net block via cross-attention (dashed lines). Intermediate features are projected into two paths: a temporal semantic branch aligned with MoCLIP features and a frequency semantic branch aligned with DCT coefficients. Both branches are modulated by timestep embeddings via FiLM and integrated with the main U-Net stream to guide denoising.

## 3 METHOD

### 3.1 PRELIMINARIES

#### 3.1.1 TEXT-TO-MOTION DIFFUSION BASICS.

Let a natural-language prompt $\mathcal{T}$ be embedded by a language encoder into a vector $\mathbf{c} = \text{CLIP}(\mathcal{T}) \in \mathbb{R}^{d_c}$. The ground-truth motion $\mathbf{x}_0 \in \mathbb{R}^{N \times d_m}$ contains $N$ frames, each parameterised by $d_m$ joints or 6-DoF pose coefficients. Following the DDPM formulation adopted by StableMoFusion Huang et al. (2024), we corrupt $\mathbf{x}_0$ with a variance schedule $\{\beta_t\}_{t=1}^{T}$:

$$\mathbf{x}_t = \sqrt{\alpha_t}\,\mathbf{x}_0 + \sqrt{1-\alpha_t}\,\boldsymbol{\epsilon}, \quad \boldsymbol{\epsilon} \sim \mathcal{N}(\mathbf{0}, \mathbf{I}), \alpha_t = \prod_{i=1}^{t}(1-\beta_i) \tag{1}$$

A UNet-style denoiser $G_\theta$ is trained to predict $\mathbf{x}_0$ (or, equivalently, the added noise) from $\mathbf{x}_t$ given the timestep $t$ and the text embedding $\mathbf{c}$:

$$\min_\theta \ \mathbb{E}_{t,\mathbf{x}_0,\boldsymbol{\epsilon},\mathbf{c}} \big\| G_\theta(\mathbf{x}_t, t, \mathbf{c}) - \mathbf{x}_0 \big\|_2^2 \tag{2}$$

At inference time a sampler such as DPM-Solver++ Lu et al. (2023) iteratively denoises from pure Gaussian noise $\mathbf{x}_T \sim \mathcal{N}(\mathbf{0}, \mathbf{I})$, while *classifier-free guidance* combines the conditional and unconditional scores:

$$\begin{aligned} G_{\text{CFG}}(\mathbf{x}_t, t, \mathbf{c}) = G_\theta(\mathbf{x}_t, t, \varnothing) \\ + \omega\left[G_\theta(\mathbf{x}_t, t, \mathbf{c}) - G_\theta(\mathbf{x}_t, t, \varnothing)\right] \end{aligned} \tag{3}$$

Here, $\varnothing$ denotes the unconditional input, and $\omega$ is a scalar hyperparameter.

### 3.2 LUMA ARCHITECTURE

Building on the StableMoFusion Huang et al. (2024) backbone, we present the **Low-dimensional Unified Motion Alignment (LUMA)** framework, as illustrated in Figure 2. LUMA introduces dual semantic anchors at the final downsampling block of the UNet architecture, aiming to enhance semantic alignment and improve gradient flow in deep layers. These anchors deliver high-signal

gradients to otherwise deep, semantically sparse and low-gradient layers, thereby alleviating gradient divergence and thus achieving a more stable and efficient training process. Meanwhile, to better capture motion semantics and encode action descriptions in text, we propose a novel text-motion alignment model, **MoCLIP**, trained from scratch using contrastive learning (see details in Appendix B). Its text encoder provides token-wise embeddings that are injected into each UNet block via cross attention.

### 3.2.1 DUAL ANCHORS.

The low-dimensional feature $\mathbf{h} \in \mathbb{R}^{C \times T}$ is extracted from the final downsampling block of the UNet, which serves as the information bridge between the encoder and decoder. Here, $C$ is the channel number and $T$ is the number of motion frames. This bottleneck stage effectively aggregates global context while preserving essential temporal structures, making it an ideal location for extracting semantically rich and compact representations.

Accordingly, LUMA applies two lightweight multilayer perceptrons (MLPs) to project $\mathbf{h}$ into distinct frequency semantic and temporal semantic representations: a frequency semantic projector $P_\theta : \mathbb{R}^{C \times T} \to \mathbb{R}^{d_m \times F}$, where $d_m$ is the motion-feature dimension and $F$ is the frequency semantic anchor dimension; and a temporal semantic projector $Q_\theta : \mathbb{R}^{C \times T} \to \mathbb{R}^{D_a}$, where $D_a$ denotes the temporal semantic anchor dimension.

**Timestep-aware modulation.** To address varying gradient propagation across diffusion timesteps, where early stages have larger gradients and later stages smaller ones, we introduce Timestep-aware Modulation. For each projection, we employ Feature-wise Linear Modulation (FiLM) Perez et al. (2017) to adaptively scale and shift the features based on the current timestep. Let $\mathbf{e}(t) \in \mathbb{R}^{d_{\mathrm{emb}}}$ denote the sinusoidal embedding of $t$, where $d_{\mathrm{emb}}$ is the embedding dimension. Two FiLM networks $F_\psi : \mathbb{R}^{d_{\mathrm{emb}}} \to \mathbb{R}^{2d_m}$ and $F_{\psi'} : \mathbb{R}^{d_{\mathrm{emb}}} \to \mathbb{R}^{2D_a}$ produce scaling and shifting vectors:

$$(\boldsymbol{\gamma}_{\mathrm{f}}(t), \boldsymbol{\beta}_{\mathrm{f}}(t)) = F_\psi\left(\mathbf{e}(t)\right), \tag{4}$$

$$(\boldsymbol{\gamma}_{\mathrm{t}}(t), \boldsymbol{\beta}_{\mathrm{t}}(t)) = F_{\psi'}\left(\mathbf{e}(t)\right), \tag{5}$$

where $\boldsymbol{\gamma}_{\mathrm{f}}(t), \boldsymbol{\beta}_{\mathrm{f}}(t) \in \mathbb{R}^{d_m}$ are for the frequency semantic anchor, and $\boldsymbol{\gamma}_{\mathrm{t}}(t), \boldsymbol{\beta}_{\mathrm{t}}(t) \in \mathbb{R}^{D_a}$ are for the temporal semantic anchor. The modulated features are:

$$\tilde{\mathbf{h}}_{\mathrm{f}} = \left((1 + \boldsymbol{\gamma}_{\mathrm{f}}(t)) \odot P_\theta(\mathbf{h}) + \boldsymbol{\beta}_{\mathrm{f}}(t)\right)^\top, \tag{6}$$

$$\tilde{\mathbf{h}}_{\mathrm{t}} = (1 + \boldsymbol{\gamma}_{\mathrm{t}}(t)) \odot Q_\theta(\mathbf{h}) + \boldsymbol{\beta}_{\mathrm{t}}(t). \tag{7}$$

The superscript $\top$ permutes the last two dimensions, $[d_m, F] \to [F, d_m]$, not a matrix transpose. The operator $\odot$ denotes element-wise multiplication, with broadcasting along $F$ in the frequency branch and none in the temporal branch.

**Frequency semantic anchor.** The **frequency semantic anchor** $\mathbf{z}_{\mathrm{fre}} = \tilde{\mathbf{h}}_{\mathrm{f}} \in \mathbb{R}^{F \times d_m}$ is aligned with the first $k$ DCT coefficients of the target motion $\mathbf{x}_0 \in \mathbb{R}^{N \times d_m}$ (where $N$ is the number of frames and $d_m$ is the motion feature dimension):

$$\mathcal{L}_{\mathrm{fre}} = \|\mathbf{z}_{\mathrm{fre}} - \mathrm{DCT}_k(\mathbf{x}_0)\|_2^2, \tag{8}$$

where $\mathrm{DCT}_k(\mathbf{x}_0) \in \mathbb{R}^{F \times d_m}$ contains the first $k$ low-frequency DCT coefficients.

**Temporal semantic anchor.** The **semantic anchor** $\mathbf{z}_{\mathrm{tem}} = \tilde{\mathbf{h}}_{\mathrm{t}} \in \mathbb{R}^{D_a}$ is aligned with the output of a frozen MoCLIP motion encoder $f_{\mathrm{tem}}(\mathbf{x}_0) \in \mathbb{R}^{D_a}$ by maximizing cosine similarity:

$$\mathcal{L}_{\mathrm{tem}} = 1 - \cos\left(\mathbf{z}_{\mathrm{tem}}, f_{\mathrm{tem}}(\mathbf{x}_0)\right). \tag{9}$$

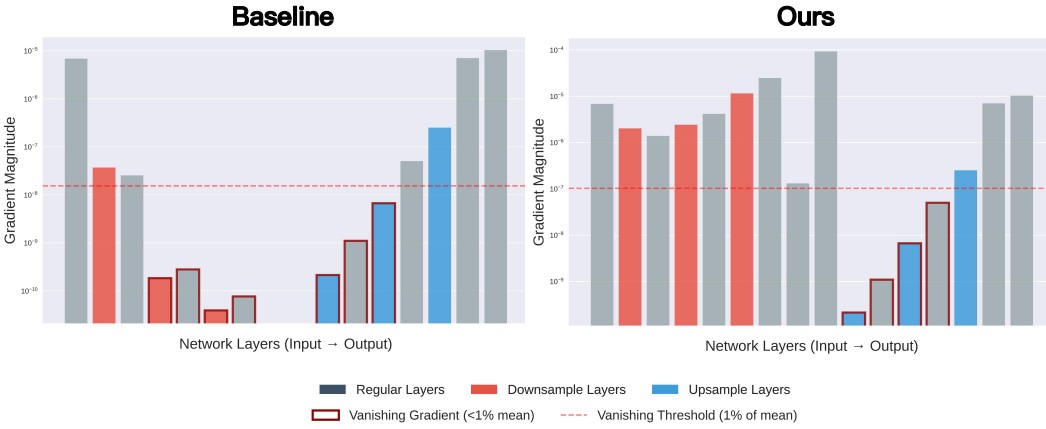

Figure 4: **Gradient magnitudes across network layers in LUMA.** Left: baseline without DAL; Right: with DAL. Red and blue bars denote downsampling and upsampling layers. The dashed line marks the vanishing gradient threshold (1% of mean). DAL boosts gradients in deep layers, mitigating vanishing issues and enhancing learning.

#### 3.2.2 DYNAMIC ANCHOR WEIGHTING.

Inspired by prior work Wang et al. (2025) advocating attenuated auxiliary supervision in later training stages, we apply a cosine annealing schedule to progressively reduce the influence of semantic alignment losses. The overall training objective integrates the DDPM reconstruction loss with the **Dual Anchor Loss (DAL)**, which combines frequency and temporal alignment losses:

$$\mathcal{L} = \mathcal{L}_{\text{DDPM}} + \zeta(n) \cdot (\lambda_{\text{fre}} \mathcal{L}_{\text{fre}} + \lambda_{\text{tem}} \mathcal{L}_{\text{tem}}), \tag{10}$$

where $\mathcal{L}_{\text{DDPM}}$ denotes the standard denoising loss, $\lambda_{\text{fre}}$ and $\lambda_{\text{tem}}$ are weighting coefficients, and $n$ is the current training step. The annealing factor $\zeta(n)$ is defined as:

$$\zeta(n) = \frac{1}{2} \left[ 1 + \cos \left( \pi \cdot \min \left( \frac{n}{N}, 1 \right) \right) \right], \tag{11}$$

with $\pi$ denoting the mathematical constant and $N$ the decay threshold. This schedule gradually suppresses DAL by step $N$, preventing over-regularization and allowing the model to refine motion generation autonomously.

## 4 EXPERIMENTS

We evaluate our approach on two standard motion-language benchmarks: HumanML3D Guo et al. (2022a) and KIT-ML Plappert et al. (2016). HumanML3D contains 14,616 motion sequences from AMASS Mahmood et al. (2019) and HumanAct12 Guo et al. (2020), each paired with three text descriptions (44,970 total), covering diverse actions like walking, exercising, and dancing. KIT-ML includes 3,911 motions and 6,278 descriptions, serving as a smaller-scale benchmark. We follow the standard evaluation protocol of StableMoFusion, including pose representation and mirroring-based augmentation. The data is split into training, validation, and test sets with a ratio of 0.8:0.15:0.05. More details about the datasets and evaluation metrics are available in **Appendix A**.

**Experimental Setup.** We adopt model architecture settings consistent with StableMoFusion Huang et al. (2024), closely following its training methodology with a total of 50,000 training steps. The diffusion process employs $T = 1000$ time steps with a linear beta schedule, and DPM-Solver is used for inference with 10 sampling steps. For DAL, $\lambda_{\text{fre}}$ is set to 0.1, and $\lambda_{\text{tem}}$ is actively used with a value of 0.5. The parameter $k$ is set to 64. In our framework, anchor selection is performed at the Down-block3 stage, where FiLM modulation is also injected. During inference, the classifier-free guidance scale is set to 2.5. A conditional mask probability of 0.1 is applied during training to enable classifier-free guidance. The entire training procedure can be efficiently completed on a single RTX 4090 GPU with 24 GB of memory.

| Method | FID ↓ | R-Precision ↑ | | | Diversity → | MM. Dist.↓ | MM. ↑ |
|---|---|---|---|---|---|---|---|
| | | Top1 | Top2 | Top3 | | | |
| **HumanML3D** | | | | | | | |
| Real | $0.002^{\pm.000}$ | $0.511^{\pm.003}$ | $0.703^{\pm.003}$ | $0.797^{\pm.002}$ | $9.503^{\pm.065}$ | - | - |
| MLD (Chen et al.) | $0.473^{\pm.013}$ | $0.481^{\pm.003}$ | $0.673^{\pm.003}$ | $0.772^{\pm.002}$ | $9.724^{\pm.082}$ | $3.196^{\pm.010}$ | $\underline{2.413}^{\pm.079}$ |
| MDM (Tevet et al.) | $0.544^{\pm.044}$ | $0.320^{\pm.005}$ | $0.498^{\pm.004}$ | $0.611^{\pm.007}$ | $\underline{9.559}^{\pm.086}$ | $5.556^{\pm.027}$ | $\mathbf{2.799}^{\pm.072}$ |
| ReMoDiffuse (Zhang et al.) | $0.103^{\pm.004}$ | $0.510^{\pm.005}$ | $0.698^{\pm.006}$ | $0.795^{\pm.004}$ | $9.018^{\pm.075}$ | $3.025^{\pm.008}$ | $1.795^{\pm.043}$ |
| T2M-GPT (Zhang et al.) | $0.141^{\pm.005}$ | $0.492^{\pm.003}$ | $0.679^{\pm.002}$ | $0.775^{\pm.002}$ | $9.722^{\pm.082}$ | $3.121^{\pm.009}$ | $1.831^{\pm.048}$ |
| MotionGPT (Jiang et al.) | $0.232^{\pm.008}$ | $0.492^{\pm.003}$ | $0.681^{\pm.003}$ | $0.778^{\pm.002}$ | $9.528^{\pm.071}$ | $3.096^{\pm.008}$ | $2.008^{\pm.084}$ |
| MotionDiffuse (Zhang et al.) | $0.630^{\pm.001}$ | $0.491^{\pm.001}$ | $0.681^{\pm.001}$ | $0.782^{\pm.001}$ | $9.410^{\pm.049}$ | $3.113^{\pm.001}$ | $1.553^{\pm.042}$ |
| MotionLCM (Dai et al.) | $0.467^{\pm.012}$ | $0.502^{\pm.003}$ | $0.701^{\pm.002}$ | $0.803^{\pm.002}$ | $9.361^{\pm.660}$ | $3.012^{\pm.007}$ | $2.172^{\pm.082}$ |
| MMM (Pinyoanuntapong et al.) | $0.089^{\pm.002}$ | $0.515^{\pm.002}$ | $0.708^{\pm.002}$ | $0.804^{\pm.002}$ | $9.577^{\pm.050}$ | $2.926^{\pm.007}$ | $1.226^{\pm.035}$ |
| MoMask (Guo et al.) | $\underline{0.045}^{\pm.002}$ | $0.521^{\pm.002}$ | $0.713^{\pm.002}$ | $0.807^{\pm.002}$ | - | $2.958^{\pm.008}$ | $1.241^{\pm.040}$ |
| StableMoFusion (Huang et al.) | $0.152^{\pm.004}$ | $\underline{0.546}^{\pm.002}$ | $\underline{0.742}^{\pm.002}$ | $\underline{0.835}^{\pm.002}$ | $\mathbf{9.466}^{\pm.002}$ | $\underline{2.781}^{\pm.011}$ | $1.362^{\pm.062}$ |
| LUMA (Ours) | $\mathbf{0.035}^{\pm.002}$ | $\mathbf{0.556}^{\pm.003}$ | $\mathbf{0.750}^{\pm.002}$ | $\mathbf{0.839}^{\pm.0013}$ | $9.596^{\pm.083}$ | $\mathbf{2.763}^{\pm.008}$ | $1.696^{\pm.049}$ |
| **KIT-ML** | | | | | | | |
| Real | $0.031^{\pm.004}$ | $0.424^{\pm.005}$ | $0.649^{\pm.006}$ | $0.779^{\pm.006}$ | $11.080^{\pm.097}$ | - | - |
| MLD (Chen et al.) | $0.404^{\pm.027}$ | $0.390^{\pm.008}$ | $0.609^{\pm.008}$ | $0.734^{\pm.007}$ | $10.800^{\pm.117}$ | $3.204^{\pm.027}$ | $\underline{2.192}^{\pm.071}$ |
| MDM (Tevet et al.) | $0.497^{\pm.021}$ | $0.164^{\pm.004}$ | $0.291^{\pm.004}$ | $0.396^{\pm.004}$ | $10.847^{\pm.119}$ | $9.191^{\pm.022}$ | $1.907^{\pm.214}$ |
| ReMoDiffuse (Zhang et al.) | $0.155^{\pm.006}$ | $0.427^{\pm.014}$ | $0.641^{\pm.004}$ | $0.765^{\pm.055}$ | $10.800^{\pm.105}$ | $2.814^{\pm.028}$ | $1.239^{\pm.028}$ |
| T2M-GPT (Zhang et al.) | $0.514^{\pm.029}$ | $0.416^{\pm.006}$ | $0.627^{\pm.006}$ | $0.745^{\pm.006}$ | $10.921^{\pm.108}$ | $3.007^{\pm.023}$ | $1.570^{\pm.039}$ |
| MotionGPT (Jiang et al.) | $0.510^{\pm.016}$ | $0.366^{\pm.005}$ | $0.558^{\pm.004}$ | $0.680^{\pm.005}$ | $10.350^{\pm.084}$ | $3.527^{\pm.021}$ | $\mathbf{2.328}^{\pm.117}$ |
| MotionDiffuse (Zhang et al.) | $1.954^{\pm.062}$ | $0.417^{\pm.004}$ | $0.621^{\pm.004}$ | $0.739^{\pm.004}$ | $\mathbf{11.100}^{\pm.143}$ | $2.958^{\pm.005}$ | $0.730^{\pm.013}$ |
| MMM (Pinyoanuntapong et al.) | $0.316^{\pm.028}$ | $0.404^{\pm.005}$ | $0.621^{\pm.005}$ | $0.744^{\pm.004}$ | $10.910^{\pm.101}$ | $2.977^{\pm.019}$ | $1.232^{\pm.039}$ |
| MoMask (Guo et al.) | $\underline{0.204}^{\pm.011}$ | $0.433^{\pm.007}$ | $0.656^{\pm.005}$ | $0.781^{\pm.005}$ | - | $2.779^{\pm.022}$ | $1.131^{\pm.043}$ |
| StableMoFusion (Huang et al.) | $0.258^{\pm.029}$ | $\underline{0.446}^{\pm.006}$ | $\underline{0.660}^{\pm.005}$ | $\underline{0.782}^{\pm.004}$ | $\underline{10.936}^{\pm.077}$ | $2.800^{\pm.018}$ | $1.362^{\pm.062}$ |
| LUMA (Ours) | $\mathbf{0.123}^{\pm.014}$ | $\mathbf{0.454}^{\pm.006}$ | $\mathbf{0.675}^{\pm.005}$ | $\mathbf{0.796}^{\pm.005}$ | $10.803^{\pm.076}$ | $\mathbf{2.711}^{\pm.015}$ | $1.233^{\pm.055}$ |

Table 1: Experimental results on HumanML3D and KIT-ML datasets. $\pm$ indicates a 95% confidence interval. **Bold** and Underline indicate the best and the second-best results, respectively. The right arrow ($\rightarrow$) denotes that a higher value is closer to real motion. Our method achieves state-of-the-art FID and R-Precision across both benchmarks.

## 4.1 COMPARISON TO STATE-OF-THE-ART APPROACHES

**Quantitative comparisons.** Following Zhang et al. (2023a); Guo et al. (2023), we report the average over 20 repeated generations with a 95% confidence interval. Table 1 presents evaluations on the HumanML3D Guo et al. (2022a) and KIT-ML Plappert et al. (2016) datasets, respectively, in comparison with state-of-the-art (SOTA) approaches.

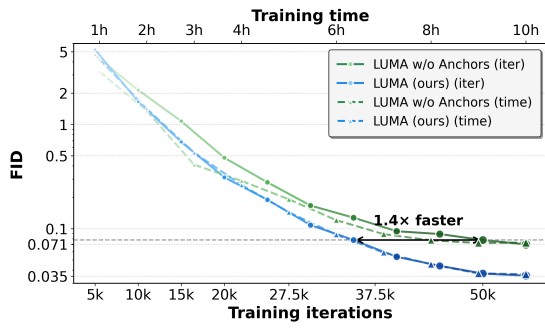

Figure 5: FID convergence curves for LUMA and the anchor-free baseline (vs. training steps and wall-clock time). The dashed line marks an FID threshold of 0.078. LUMA reaches this in **35k** iterations (**1.4×** faster), reducing training time from 8 to **7 hours**, demonstrating more efficient convergence.

In terms of empirical results, LUMA demonstrates strong performance across all key evaluation metrics. Our method achieves substantial improvements in FID (HumanML3D: **0.035**; KIT: **0.123**) and R-Precision (Top-3 HumanML3D: **0.839**; KIT: **0.796**), clearly illustrating the effectiveness of LUMA in enhancing the baseline model. Compared to state-of-the-art VQ-based approaches such as MMM Pinyoanuntapong et al. (2024c) and MoMask Guo et al. (2023), LUMA consistently achieves competitive or superior results on the HumanML3D dataset, outperforming other methods in terms of FID, R-Precision, and MultiModal Distance. This highlights LUMA's ability to generate high-quality, semantically aligned, and diverse motion sequences. On the more challenging KIT-ML benchmark, LUMA attains the best FID among all evaluated methods and ranks first in R-Precision, demonstrating the broad adaptability of the LUMA architecture. Overall, these results underscore the capability of LUMA to significantly improve both the quality and diversity of text-driven motion generation.

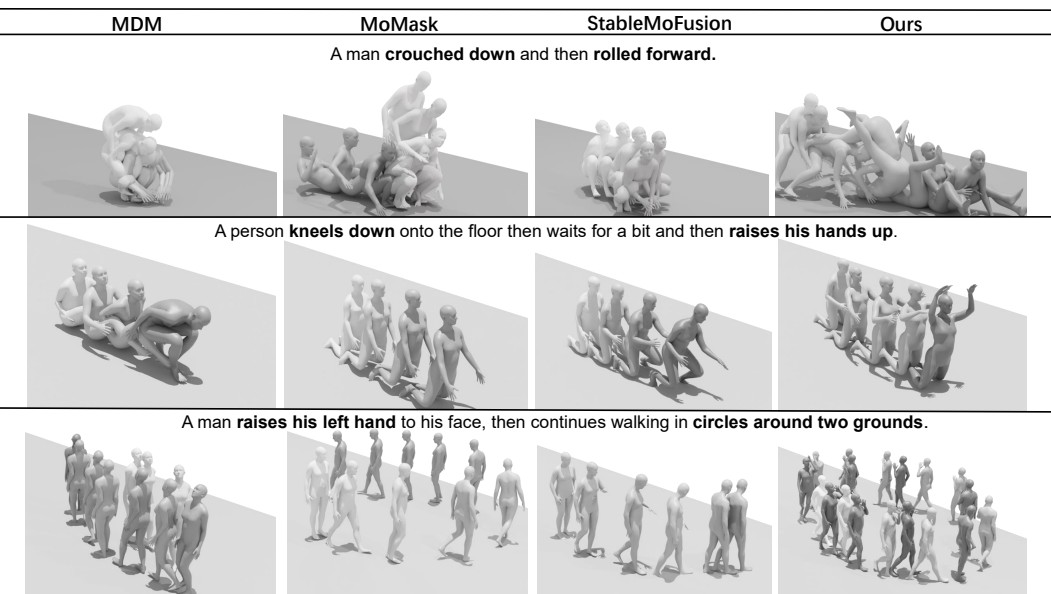

Figure 6: Visualization Comparison. We compare the visual results of LUMA with three state-of-the-art methods. In both examples, LUMA consistently produces more accurate, natural, and fine-grained motion than the other approaches.

**Qualitative comparison.** Figure 6 presents visual comparisons with MDM, StableMoFusion, and MoMask under three prompts. For "A man crouched down and then rolled forward," MDM and StableMoFusion only crouch, while MoMask fails to roll properly. Our model performs the full action accurately. For "A person kneels down onto the floor then waits and raises his hands up," MDM misses kneeling, and others raise hands poorly; our method captures both with correct timing. For the complex prompt involving hand-raising and circling, baselines fail to complete all sub-actions, while our model executes them with high fidelity. These results demonstrate superior semantic understanding, temporal consistency, and detail accuracy.

## 4.2 FASTER CONVERGENCE WITH DUAL ANCHORS

As shown in Figure 5, LUMA achieves significantly faster convergence compared to the anchor-free baseline. Our model surpasses the FID threshold of 0.078 with only **35k** training iterations, whereas the baseline requires **50k** steps, resulting in a **1.4×** speedup. This efficiency gain is also reflected in the wall-clock training time. As illustrated on the upper axis of the same figure, LUMA reaches the target FID in just **7 hours**, while the baseline takes **8 hours**. This demonstrates that our dual-anchor alignment not only improves motion quality but also makes training substantially more efficient, both in computation steps and real-world time.

## 4.3 GRADIENT ANALYSIS

As shown in Figure 4, removing the DAL causes the gradient magnitudes in the middle layers to drop sharply. The downsampling path is especially affected. Some layers fall below the vanishing threshold (1% of the mean). This indicates severe gradient attenuation in these regions. After introducing the DAL, the gradients in the deep downsampling layers increase by nearly two orders of magnitude and are brought to a level comparable with the upsampling path. This observation validates our earlier hypothesis that vanishing gradients in deep layers hinder effective learning of abstract features. The additional semantic and motion anchors introduced by DAL provide more informative supervisory signals to the encoder, substantially alleviating the vanishing gradient problem and accelerating effective representation learning in deeper layers.

| Design Choice | FID $\downarrow$ | R@3 $\uparrow$ |
|---|---|---|
| DCT $k = 8$ | $0.044^{\pm.003}$ | $0.835^{\pm.003}$ |
| DCT $k = 16$ | $0.052^{\pm.002}$ | $0.829^{\pm.003}$ |
| DCT $k = 32$ | $0.052^{\pm.004}$ | $0.832^{\pm.002}$ |
| **DCT $k = 64$** | $\mathbf{0.035}^{\pm.002}$ | $\mathbf{0.839}^{\pm.001}$ |
| DCT $k = 128$ | $0.063^{\pm.004}$ | $0.831^{\pm.003}$ |
| Down-block1 | $0.103^{\pm.003}$ | $0.813^{\pm.002}$ |
| Down-block2 | $0.076^{\pm.001}$ | $0.827^{\pm.001}$ |
| **Down-block3** | $\mathbf{0.035}^{\pm.002}$ | $\mathbf{0.839}^{\pm.001}$ |
| Bottleneck | $0.092^{\pm.002}$ | $0.838^{\pm.003}$ |

Table 3: Ablation study on key design choices in LUMA. For each factor, the best configuration is highlighted in bold.

| Anchor Weighting Strategy | FID $\downarrow$ | R@3 $\uparrow$ |
|---|---|---|
| Static (fixed) | $0.047^{\pm.001}$ | $0.837^{\pm.002}$ |
| Learnable (global) | $0.051^{\pm.002}$ | $0.834^{\pm.002}$ |
| **Dynamic (cosine)** | $\mathbf{0.035}^{\pm.002}$ | $\mathbf{0.839}^{\pm.001}$ |

Table 4: Ablation on anchor weighting strategies. The dynamic schedule achieves the best performance.

## 4.4 ABLATION STUDY

To comprehensively assess the contribution of each component in our framework, we conduct a series of controlled ablation experiments. Following the evaluation protocol, we focus on two aspects: (i) the effect of each core module, (ii) the influence of key design choices. All experiments are performed under identical training settings for fair comparison.

**Core Component Analysis.** As shown in Table 2, each component of LUMA plays a critical and complementary role. The backbone alone achieves a baseline FID of 0.152, while removing both anchors but retaining the LUMA architecture already yields a substantial FID drop to 0.067, highlighting structural benefits. Introducing either the frequency semantic or temporal semantic anchor alone further reduces FID to 0.057 and improves R@3 to 0.840, indicating that both anchors independently enhance motion quality and text alignment. Their similar contributions suggest that each focuses on a different aspect of the generation process. One captures frequency-level structure, while the other improves semantic consistency. Removing MoCLIP leads to a slight drop in performance, confirming its importance in providing effective semantic supervision. Meanwhile, eliminating FiLM leads to a marginal drop in R@3 despite strong FID, showing its importance in temporal modulation. The full model outperforms all variants (FID 0.035, R@3 0.839), demonstrating that these modules act synergistically to improve fidelity, semantic alignment, and training stability.

**Design Choice Exploration.** We study two key hyperparameters in LUMA: the number of retained DCT coefficients ($k$) and the FiLM injection location. As shown in Table 3, an intermediate setting ($k = 64$) best balances global structure and noise suppression; smaller ($k = 8$) limits expressiveness, while larger ($k = 128$) introduces artifacts. For FiLM, injecting at *Down-block3* provides the best trade-off, capturing sufficient temporal context without losing fine details or adding notable compute.

| Method | FID $\downarrow$ | R@3 $\uparrow$ |
|---|---|---|
| StableMoFusion (backbone) | $0.152^{\pm.004}$ | $0.835^{\pm.002}$ |
| LUMA w/o DAL | $0.067^{\pm.003}$ | $0.835^{\pm.002}$ |
| LUMA w/o $\mathcal{L}_{\text{tem}}$ | $0.057^{\pm.002}$ | $0.840^{\pm.002}$ |
| LUMA w/o $\mathcal{L}_{\text{fre}}$ | $0.057^{\pm.002}$ | $0.840^{\pm.002}$ |
| LUMA w/o MoCLIP | $0.075^{\pm.003}$ | $0.838^{\pm.002}$ |
| LUMA w/o FiLM | $0.056^{\pm.003}$ | $0.833^{\pm.002}$ |
| **LUMA (full)** | $\mathbf{0.035}^{\pm.002}$ | $\mathbf{0.839}^{\pm.001}$ |

Table 2: Ablation results for core components of LUMA on the main benchmark. We report FID ($\downarrow$) and R@3 ($\uparrow$) with mean$\pm$95% CI.

**Dynamic Anchor Weighting.** We compare
our dynamic cosine-annealed schedule with
(i) *static averaging* and (ii) *fixed learnable weights*. Table 4 shows the dynamic scheme consistently
achieves the best FID and R@3. It emphasizes temporal and frequency alignment early, then gradually relaxes constraints to avoid over-regularization, enabling finer motion refinement and stronger
semantic consistency.

## 5 CONCLUSION

In conclusion, we identify severe gradient sparsity in deep layers of diffusion-based motion models,
which limits semantic and structural alignment. To address this, we propose **LUMA**, a dual-path
framework injecting temporal and frequency anchors into the diffusion backbone. This design enhances gradient flow, speeds up convergence, and improves motion fidelity. Experiments on HumanML3D and KIT-ML show state-of-the-art FID and R-Precision, with strong diversity and multimodality. Ablations and gradient analyses further validate the effectiveness of our approach and
support our hypothesis.

## ETHICS STATEMENT

This work focuses on advancing text-to-motion generation models. The proposed methods and experiments do not involve sensitive data, personally identifiable information, or human subjects. The potential societal impact lies primarily in creative applications such as animation, gaming, VR/AR, and robotics. We acknowledge the importance of responsible use and encourage future work to mitigate potential misuse in areas such as generating misleading or harmful content.

## REPRODUCIBILITY STATEMENT

We have taken extensive steps to ensure reproducibility. Details of the model architecture, hyper-parameters, datasets, and training procedure are provided in the main text and the Appendix. We use the publicly available HumanML3D and KIT-ML datasets. To further facilitate replication, we include the full source code, training scripts, and pre-trained models in the Supplementary Material, enabling direct reproduction of our results. Our experiments were performed with clearly specified hardware and software configurations, ensuring that other researchers can reliably replicate and extend our work.

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

APPENDIX

# A    MOTION DATA REPRESENTATION AND EVALUATION METRICS

To comprehensively evaluate text-to-motion generation, we utilize a suite of established quantitative metrics and adopt precise motion data representations. This enables rigorous analysis of both generation performance and data fidelity. We first introduce the evaluation metrics, followed by a detailed discussion of motion representations.

## A.1    EVALUATION METRICS

We employ widely recognized quantitative metrics including Frechet Inception Distance (FID) Heusel et al. (2017), R-Precision, Diversity, Multimodal Distance (MM-Dist.), and Multimodality (MM.), consistent with the evaluation protocol of StableMoFusion Huang et al. (2024). Additionally, human evaluation is conducted to assess semantic accuracy and human preference, providing a supplementary subjective perspective.

- **Frechet Inception Distance (FID):** Measures the distributional difference between generated and real motion features, reflecting the overall motion quality.
- **R-Precision:** Evaluates motion-to-text retrieval by ranking Euclidean distances between each generated motion and 32 candidate text descriptions (1 ground-truth, 31 mismatched). Top-1, Top-2, and Top-3 retrieval accuracies are reported.
- **Diversity:** Quantifies motion diversity by averaging Euclidean distances between 300 randomly sampled pairs of generated motions.
- **Multimodal Distance (MM-Dist.):** Assesses the semantic alignment between generated motions and corresponding texts. Lower MM-Dist. indicates better cross-modal correspondence.
- **Multimodality (MM.):** Measures the diversity of motions generated from a single text prompt. For each prompt, 20 motions are generated to form 10 pairs; the mean Euclidean distance between pairs is computed and averaged across all prompts.

## A.2    MOTION DATA REPRESENTATIONS

We analyze two predominant motion representation formats: the *HumanML3D Format* and the *SMPL-based Format* Loper et al. (2015). Both are widely adopted in prior works and offer complementary advantages in semantic expressiveness and biomechanical fidelity.

**HumanML3D Format.**    The HumanML3D format encodes motion as a tuple of spatial and dynamic features:

$$x^i = \{r^a, r^x, r^z, r^y, j^P, j^v, j^r, \mathbf{f}_{\text{contact}}\}, \tag{12}$$

where $r^a$ is the root angular velocity (Y-axis), $r^x$ and $r^z$ are root linear velocities (XZ-plane), $r^y$ is root height, $j^P \in \mathbb{R}^{3N_j}$ denotes local joint positions, $j^v \in \mathbb{R}^{3N_j}$ and $j^r \in \mathbb{R}^{6N_j}$ represent joint velocities and joint rotations, and $\mathbf{f}_{\text{contact}} \in \mathbb{R}^4$ are binary foot-ground contact features. This comprehensive definition captures both high-level semantic structure and fine-grained motion detail.

**SMPL-based Format.**    The SMPL model focuses on anatomical accuracy. Each motion is represented as:

$$x^i = \{r, \boldsymbol{\vartheta}, \beta\}, \tag{13}$$

where $r$ denotes global translation, $\boldsymbol{\vartheta} \in \mathbb{R}^{3 \times 23+3}$ represents joint rotations for 23 joints and a root joint, and $\beta$ encodes body shape parameters. The SMPL-based representation is well-suited for modeling biomechanically realistic motions.

**Frequency-Domain Representation.**    In addition to these established formats, we further introduce a novel frequency-domain representation to enhance semantic planning and detailed motion refinement, bridging global temporal structure with local motion accuracy.

## B    DETAILS OF MoCLIP

Inspired by CLIP's image-text alignment success MoCLIP extends this concept to human motion. It learns a shared embedding space mapping textual descriptions to corresponding motion sequences enabling cross-modal retrieval and understanding.

### B.0.1    ARCHITECTURE

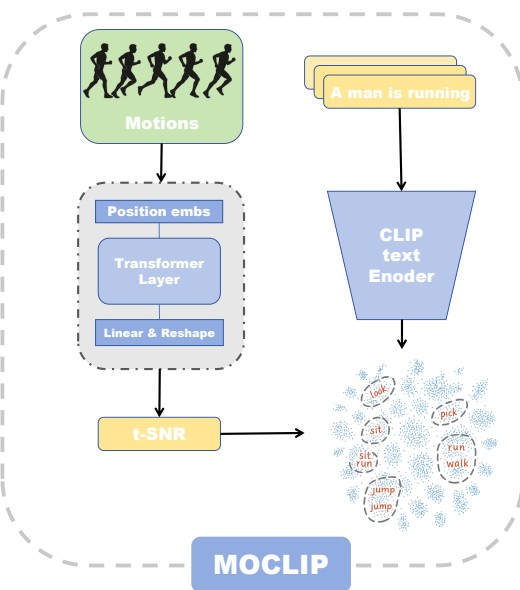

As shown in Figure 7, MoCLIP employs a dual-encoder structure adapting the pretrained CLIP architecture. The text pathway uses a fine-tuned CLIP text encoder generating semantic embeddings $\mathbf{z}_{\text{text}} \in \mathbb{R}^{d_{\text{MoCLIP}}}$. The motion pathway features a dedicated MotionEncoder processing sequences $\mathbf{M} \in \mathbb{R}^{N_{\text{seq}} \times d_{\text{motion}}}$. This encoder applies linear projection $\mathbf{W}_p$ adds sinusoidal positional encoding $\mathbf{P}$ passes the result through $L$ transformer layers handling variable lengths via mask $\mathbf{M}_{mask}$ performs temporal average pooling and finally projects features via $\mathbf{W}_o$ to obtain motion embeddings $\mathbf{z}_{\text{motion}} \in \mathbb{R}^{d_{\text{MoCLIP}}}$. The core operation within the transformer layers is multi-head self-attention:

$$\mathbf{s}_l = \text{TransformerLayer}(\mathbf{s}_{l-1}, \mathbf{M}_{mask}). \quad (14)$$

Contrastive learning in the shared $d_{\text{MoCLIP}}$-dimensional space aligns these embeddings scaled by temperature $\tau$.

### B.0.2    LOSS FUNCTION

MoCLIP utilizes a symmetric contrastive loss to align modalities:

$$\mathcal{L}_{\text{contrastive}} = \frac{1}{2}\left(\mathcal{L}_{\text{motion-to-text}} + \mathcal{L}_{\text{text-to-motion}}\right). \quad (15)$$

Figure 7: **Overview of MoCLIP.** MoCLIP aligns motion and text through contrastive learning, using a transformer-based motion encoder and a CLIP text encoder to build a shared semantic space.

This loss averages the motion-to-text and text-to-motion cross-entropy terms. These terms are computed using cosine similarity between L2-normalized motion and text embeddings promoting high similarity for matched pairs and low similarity for mismatched pairs.

### B.0.3    TRAINING STRATEGY

Training follows a two-stage strategy for progressive alignment. **Stage 1: Motion Encoder Pre-training.** The CLIP text encoder is frozen. Only the MotionEncoder components are trained optimizing the contrastive loss (Eq. 15). This initially aligns motion features to the fixed text embedding space. **Stage 2: Joint Fine-Tuning.** The final layers of the CLIP text encoder are unfrozen. The entire model is then fine-tuned jointly with a lower learning rate. This allows mutual refinement of both motion and text representations enhancing the joint embedding space. This approach facilitates stable learning and effective cross-modal integration.

### B.1    PERFORMANCE OF MoCLIP

We evaluated MoCLIP's core text-motion alignment capability on the HumanML3D Guo et al. (2022a) KIT Plappert et al. (2016) and CMP Liao et al. (2024) (Combat Motion Processed) datasets. Performance was measured using standard Top-$k$ retrieval accuracy (*Top-1 Top-2 Top-3*). Table 5 shows MoCLIP significantly outperforms the baseline across all datasets. Notably on HumanML3D MoCLIP achieves 0.705 Top-1 accuracy versus the baseline's 0.511. On CMP the improvement is also substantial reaching 0.748 Top-1 accuracy compared to 0.335. Consistent gains are observed on the KIT dataset. These results validate MoCLIP's effectiveness in learning accurate text-motion semantic mappings.

|          | Dataset   | Top-1 | Top-2 | Top-3 |
|----------|-----------|-------|-------|-------|
| Baseline | Humanml3d | 0.511 | 0.703 | 0.797 |
| MoCLIP   | Humanml3d | 0.705 | 0.856 | 0.913 |
| Baseline | KIT       | 0.424 | 0.649 | 0.779 |
| MoCLIP   | KIT       | 0.469 | 0.676 | 0.788 |
| Baseline | CMP       | 0.335 | 0.513 | 0.628 |
| MoCLIP   | CMP       | 0.748 | 0.891 | 0.942 |

Table 5: Top-$k$ retrieval accuracy comparison between the baseline and MoCLIP on HumanML3D, KIT, and CMP datasets.

| Method | FID $\downarrow$ | R-Precision $\uparrow$ | | |
|--------|------------------|------|------|------|
|        |                  | top1 | top2 | top3 |
| MDM baseline | $0.544^{\pm.044}$ | $0.320^{\pm.005}$ | $0.498^{\pm.004}$ | $0.611^{\pm.007}$ |
| MDM MoCLIP | $0.527^{\pm.034}$ | $0.514^{\pm.003}$ | $0.719^{\pm.001}$ | $0.820^{\pm.001}$ |
| Momask baseline | $0.045^{\pm.002}$ | $0.521^{\pm.002}$ | $0.713^{\pm.002}$ | $0.807^{\pm.002}$ |
| Momask MoCLIP | $0.065^{\pm.002}$ | $0.529^{\pm.002}$ | $0.724^{\pm0.002}$ | $0.818^{\pm.002}$ |
| StableMoFusion baseline | $0.152^{\pm.004}$ | $0.546^{\pm.002}$ | $0.742^{\pm.002}$ | $0.835^{\pm.002}$ |
| StableMoFusion MoCLIP | $0.067^{\pm.003}$ | $0.549^{\pm.002}$ | $0.742^{\pm.002}$ | $0.835^{\pm.002}$ |

Table 6: Evaluation results of MoCLIP integration with different models. The table shows the FID (lower is better) and R-Precision (higher is better) at top1, top2, and top3 for MDM, MoMask, and StableMoFusion models with and without MoCLIP. The results demonstrate the positive impact of MoCLIP on improving both FID and R-Precision scores in motion generation tasks.

## B.2 INTEGRATING MOCLIP FOR ENHANCED GENERATION

To evaluate the practical benefit of MoCLIP's learned representations we integrated its fine-tuned text encoder $\mathcal{T}_{\text{MoCLIP}}$ into existing generation frameworks. Specifically we replaced the native text encoders of StableMoFusion MDM and MoMask with $\mathcal{T}_{\text{MoCLIP}}$. This modification supplies these generators with motion-aligned text embeddings $\mathbf{z}_{\text{text}}^{\text{MoCLIP}}$ leveraging the shared semantic space detailed in Section B. The resulting performance improvements detailed in Table 6 demonstrate the efficacy of this approach. Using $\mathcal{T}_{\text{MoCLIP}}$ consistently enhances generation quality across the tested models. This confirms that the MoCLIP encoder effectively extracts motion-relevant semantics transforming text descriptions into representations more conducive to high-fidelity motion synthesis.

To investigate the impact of our motion-text alignment training, we performed a comparative analysis of the attention mechanisms within the text encoders of MoCLIP and the original CLIP model. Our analysis focuses on the attention maps produced by the final Transformer layer. As shown in Figure 11, it is crucial to note that at this terminal stage of encoding, the initial, discrete token embeddings have been iteratively transformed into highly contextualized feature vectors, where each vector at a given position encapsulates rich, sequence-wide semantic information. Our methodology involved computing the total attention received by the feature vector at each sequence position from all other positions, a value which we then averaged across all attention heads and normalized. By differencing these final-layer attention distributions (MoCLIP vs. original CLIP), we uncovered a systematic and significant reallocation of attention. Specifically, MoCLIP learns to assign substantially greater attentional weight to the feature vectors located at positions that correspond to the initial action-oriented tokens (e.g., verbs like "crouched", "rolled", "waves" and adverbs of motion). This strategic focus indicates that our contrastive training paradigm has successfully guided the model to identify the semantic loci of action within the sequence and to prioritize these now-contextualized features when constructing the final, motion-aware text representation. This learned

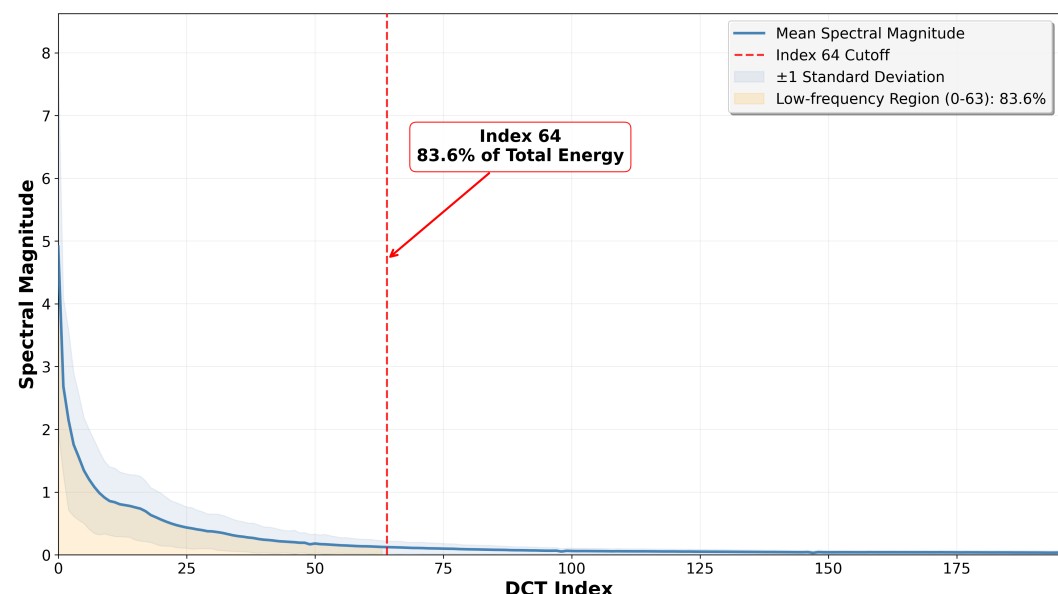

Figure 8: Mean DCT spectrum of HumanML3D. The first 64 coefficients (shaded) account for 83.6% of the total energy.

ability to pinpoint and amplify motion-centric semantics is fundamental to MoCLIP's superior performance in translating textual descriptions into complex human motion.

## C  PSEUDO-CODE

Algorithm 1 shows LUMA's Pseudo-code.

---

**Algorithm 1:** Concise LUMA Training Loop

**Input:** Dataset $\mathcal{D}$; frozen MoCLIP encoders $f_{\text{text}}, f_{\text{tem}}$;
model params $\Theta = \{\phi, \theta, \psi, \psi'\}$; hyper-params $(k, \lambda_{\text{fre}}, \lambda_{\text{tem}}, N_{\text{decay}}, T)$
**Output:** Updated $\Theta$

1 **for** $n = 1$ **to** $N_{train}$ **do**
    // Diffuse a clean sample
2    Sample $(\mathbf{x}_0, \mathbf{c})$,   $t \sim \mathcal{U}[1, T]$,   $\boldsymbol{\epsilon} \sim \mathcal{N}(0, I)$;
3    $\mathbf{x}_t \leftarrow \sqrt{\alpha_t}\mathbf{x}_0 + \sqrt{1 - \alpha_t}\boldsymbol{\epsilon}$;
    // Predict noise and extract bottleneck
4    $(\hat{\boldsymbol{\epsilon}}, \mathbf{h}) \leftarrow \text{DENOISER}_\phi(\mathbf{x}_t, f_{\text{text}}(\mathbf{c}), t)$;
    // Dual-anchor projection and FiLM modulation
5    $(\mathbf{z}_{\text{fre}}, \mathbf{z}_{\text{tem}}) \leftarrow \text{ANCHOR}_{\theta,\psi,\psi'}(\mathbf{h}, t)$;
    // Compute total loss
6    $\zeta \leftarrow \frac{1}{2}\left[1 + \cos\left(\pi \min\left(\frac{n}{N_{\text{decay}}}, 1\right)\right)\right]$;
7    $\mathcal{L}_{\text{total}} \leftarrow \underbrace{\|\hat{\boldsymbol{\epsilon}} - \boldsymbol{\epsilon}\|_2^2}_{\mathcal{L}_{\text{DDPM}}} + \zeta\left(\lambda_{\text{fre}}\|\mathbf{z}_{\text{fre}} - \text{DCT}_k(\mathbf{x}_0)\|_2^2 + \lambda_{\text{tem}}\left(1 - \cos(\mathbf{z}_{\text{tem}}, f_{\text{tem}}(\mathbf{x}_0))\right)\right)$;
    // SGD / Adam optimization step
8    $\Theta \leftarrow \Theta - \eta\nabla_\Theta\mathcal{L}_{\text{total}}$;

---

## D  HYPERPARAMETER SEARCH RESULT

In our study, we conducted a grid search on the validation set to determine the optimal weights for the dual-anchor losses in LUMA, specifically $\lambda_{\text{fre}}$ (frequency anchor) and $\lambda_{\text{tem}}$ (temporal anchor). We selected five candidate values for each hyperparameter, resulting in 25 combinations, with ranges extended beyond prior work to more fully explore the parameter space. Each combination was evaluated five times to ensure the reliability of our results.

Our analysis revealed an inverse relationship between FID and top-k accuracy when either anchor weight was set to an extreme value: lower FID often came at the expense of top-k accuracy, and vice versa. However, for most mid-range values, performance remained robust and stable, indicating that LUMA is not overly sensitive to moderate changes in these hyperparameters.

Based on these results, we selected $\lambda_{\text{fre}} = 0.10$ and $\lambda_{\text{tem}} = 0.50$ as our final configuration, achieving an optimal trade-off between perceptual quality (FID) and semantic alignment (top-k metrics) for all subsequent experiments.

## E  DETAILS OF HUMAN EVALUATION

**Task design.**  We utilize the Google Form platform to conduct a human evaluation study involving 20 independent participants. For each test prompt, we generate *four* motion clips: one from our model and three from baseline methods MDM, MoMask, StableMoFusion, denoted as $\mathcal{B}_1, \mathcal{B}_2, \mathcal{B}_3$. Each trial presents one baseline clip and the corresponding clip generated by our model, displayed side-by-side. The identity of the baseline is concealed, and the left/right order is randomized. Each participant is shown 100 such trials, covering different prompts and baseline combinations, and is asked to answer three questions:

---

**Human-study questionnaire**

**Motion A** (left) and **Motion B** (right) are two candidates for the same text prompt.

1. **Q1: Semantic accuracy.**

   Is Motion A semantically accurate?
   (a) Yes
   (b) No

   Is Motion B semantically accurate?
   (a) Yes
   (b) No

2. **Q2: Complex-action completion.**

   Is the complex movement well completed in Motion A?
   (a) Yes
   (b) No

   Is the complex movement well completed in Motion B?
   (a) Yes
   (b) No

3. **Q3: Preference.**

   Which result do you prefer?
   (a) Motion A
   (b) Motion B

---

**Metric Computation.**  We report three metrics for our model and each baseline, computed as follows:

1. **Semantic Accuracy (%):** The proportion of trials where the motion is rated as semantically consistent with the input text (Q1).

2. **Complex-Movement Completion (%):** The proportion of trials where the motion is judged to have successfully completed the described complex action (Q2).

3. **User Preference (%):** The proportion of pairwise comparisons in which our model is preferred over a given baseline (Q3).

| Method | Sem. Acc. | Compl. Comp. | Pref. |
|---|---|---|---|
| MDM | 55.3% | 52.7% | 84.1% |
| MoMask | 70.8% | 72.4% | 75.6% |
| StableMoFusion | 60.2% | 55.5% | 80.9% |
| **Ours** | **85.7%** | **86.3%** | – |

Table 7: Human evaluation results (all values in %). "Sem. Acc." is Semantic Accuracy, "Compl. Comp." is Complex-Movement Completion, and "Pref." is User Preference.

**Results.** Ours achieves the highest semantic accuracy (85.7%) and complex-action completion (86.3%); in pairwise comparisons annotators preferred our results over each baseline in 75.6–84.1% of trials, confirming both objective and subjective superiority while keeping the evaluation logic transparent and reproducible.

## F   DISCRETE COSINE TRANSFORM ANALYSIS

### F.0.1   DEFINITION

For a discrete signal $\mathbf{v} = \{v[0], v[1], \ldots, v[N_{\text{sig}} - 1]\}$, its type-II Discrete Cosine Transform (DCT) is

$$v_f[k] = A(k) \sum_{n=0}^{N_{\text{sig}}-1} v[n] \cos\left(\frac{\pi(2n+1)k}{2N_{\text{sig}}}\right), \quad k = 0, \ldots, N_{\text{sig}} - 1, \tag{16}$$

where the normalisation factor is

$$A(k) = \begin{cases} \sqrt{\frac{1}{N_{\text{sig}}}}, & k = 0, \\ \sqrt{\frac{2}{N_{\text{sig}}}}, & k > 0. \end{cases} \tag{17}$$

### F.0.2   FREQUENCY ANALYSIS ON HUMANML3D

Figure 8 shows the average spectral magnitude of HumanML3D motions after applying the DCT equation 16. We observe that the signal energy is highly concentrated in the low-frequency region: the first 64 DCT coefficients, comprising only one-third of the total frequency range, capture 83.6% of the total spectral energy. These low-frequency components primarily encode the smooth, semantically meaningful motion trajectories, while higher frequencies contribute mainly to fine-grained details and noise.

Motivated by this empirical evidence, LUMA sets the frequency semantic anchor dimension to $k = 64$, ensuring that the model retains the most informative, energy-rich motion patterns while discarding redundant high-frequency noise and reducing computational cost.

## G   GRADIENT-FLOW ANALYSIS

To quantify optimisation dynamics we attach hooks to every UNet block and record gradient magnitudes after 40 k training steps on HumanML3D. As Figure 10 shows, the baseline exhibits a spatial profile *U-shaped*: the encoder and decoder peripheries maintain healthy gradients, while the bottleneck layers suffer from one to two orders of magnitude attenuation, sometimes dipping below the 1 % threshold, effectively failing to learn. Temporally, the profile is *non-monotonic*: gradients in deep

| Method | FID ↓ | R-Precision ↑ | | |
|--------|-------|------|------|------|
| | | top1 | top2 | top3 |
| StableMoFusion | $0.291^{\pm.008}$ | $0.427^{\pm.002}$ | $0.603^{\pm.003}$ | $0.700^{\pm.003}$ |
| **LUMA** | $\mathbf{0.135}^{\pm.010}$ | $\mathbf{0.430}^{\pm.003}$ | $\mathbf{0.606}^{\pm.004}$ | $\mathbf{0.703}^{\pm.004}$ |

Table 8: Comparison on synonym-perturbed text. LUMA achieves better FID and R-Precision, indicating higher robustness.

| $\lambda_{\mathrm{fre}}\backslash\lambda_{\mathrm{tem}}$ | **0.05** | **0.10** | **0.20** | **0.35** | **0.50** |
|------|------|------|------|------|------|
| **0.05** | 0.5461/0.059 | 0.5524/0.042 | 0.5448/0.041 | 0.5379/0.036 | 0.5412/0.037 |
| **0.10** | 0.5397/0.054 | 0.5563/0.045 | 0.5481/0.049 | 0.5422/0.048 | **0.5556/0.035** |
| **0.20** | 0.5486/0.056 | 0.5619/0.044 | 0.5514/0.052 | 0.5457/0.050 | 0.5542/0.039 |
| **0.35** | 0.5449/0.068 | 0.5417/0.040 | 0.5485/0.047 | 0.5433/0.034 | 0.5396/0.036 |
| **0.50** | 0.5504/0.046 | 0.5557/0.049 | 0.5467/0.048 | 0.5479/0.038 | 0.5508/0.043 |

Table 9: LUMA dual-anchor grid search. Each cell shows Top-1 Accuracy ↑ / FID ↓. Anchor weights $\lambda_{\mathrm{fre}}$ and $\lambda_{\mathrm{tem}}$ vary in $\{0.05, 0.10, 0.20, 0.35, 0.50\}$.

layers are largest at early timesteps ($t$=750) when the network must infer global, abstract motion structure from noise, but they decay rapidly as $t \to 0$, indicating that late denoising relies chiefly on shallow layers to polish local details. This spatiotemporal imbalance both motivates our *Timestep-aware FiLM modulation*, which reinjects semantic signals proportional to $t$ and justifies the design of *Dual Anchor Loss*, whose intermediate supervision revitalises the vanishing bottleneck gradients.

# H    ROBUSTNESS ANALYSIS

To evaluate the robustness of our LUMA model under real-world language variations, we conducted experiments on a synonym-perturbed dataset constructed following the protocol of SATO Chen et al. (2024) , where the textual descriptions are randomly replaced with contextually appropriate synonyms. The quantitative results (Table 8) demonstrate that LUMA consistently outperforms the baseline StableMoFusion across all key metrics. Notably, LUMA achieves a substantially lower FID (0.135 vs. 0.291), indicating that the generated motions maintain higher fidelity to the ground truth even when the input text is perturbed. Similarly, LUMA yields higher R-Precision scores at top-1, top-2, and top-3 ranks, reflecting a stronger alignment between the generated motions and the intended semantic content of the perturbed prompts. These results confirm that LUMA is significantly more robust to synonym-level variations in natural language input, thereby improving the reliability and practicality of text-to-motion generation in real-world, linguistically diverse scenarios.

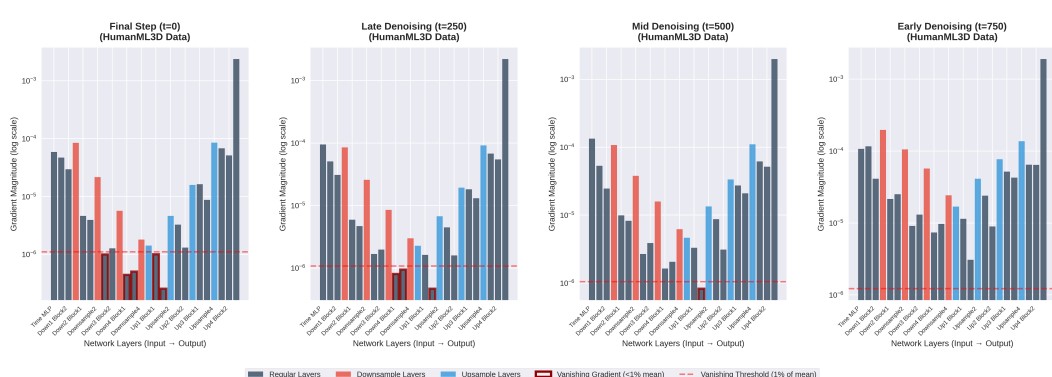

Figure 9: **Gradient magnitude across UNet layers (baseline, no DAL).** At all timesteps ($t \in \{750, 500, 250, 0\}$), gradients form a U-shaped spatial profile, with severe attenuation in deep bottleneck layers. As $t$ decreases, the gradient further weakens, suggesting spatio-temporal specialization during denoising.

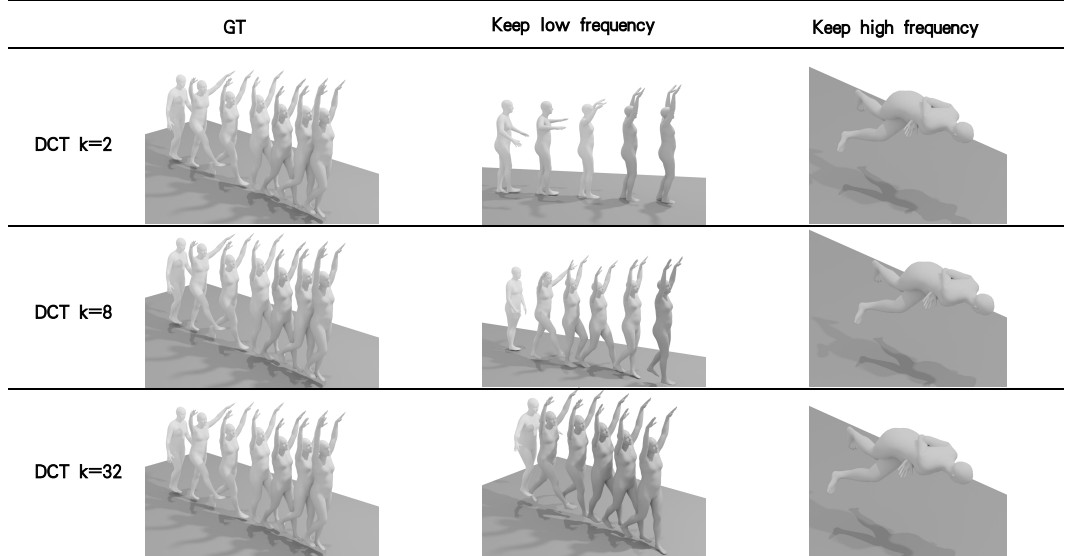

Figure 10: Visualization of motion reconstruction under DCT filtering for the text prompt *"a man walks forward raising his hands"*. Each row corresponds to a different cutoff value $K \in \{2, 8, 32\}$. The middle column ("Keep low frequency") retains only the first $K$ low-frequency DCT coefficients, while the right column ("Keep high frequency") keeps only the last $K$ high-frequency coefficients. Keeping low-frequency components preserves the global trajectory and the core semantic structure of the action, and fidelity improves as $K$ increases. In contrast, retaining only high-frequency components removes semantic content and leads to fragmented, unstable motion. These results indicate that the linguistic and semantic meaning of human motion is primarily carried by low-frequency DCT bands.

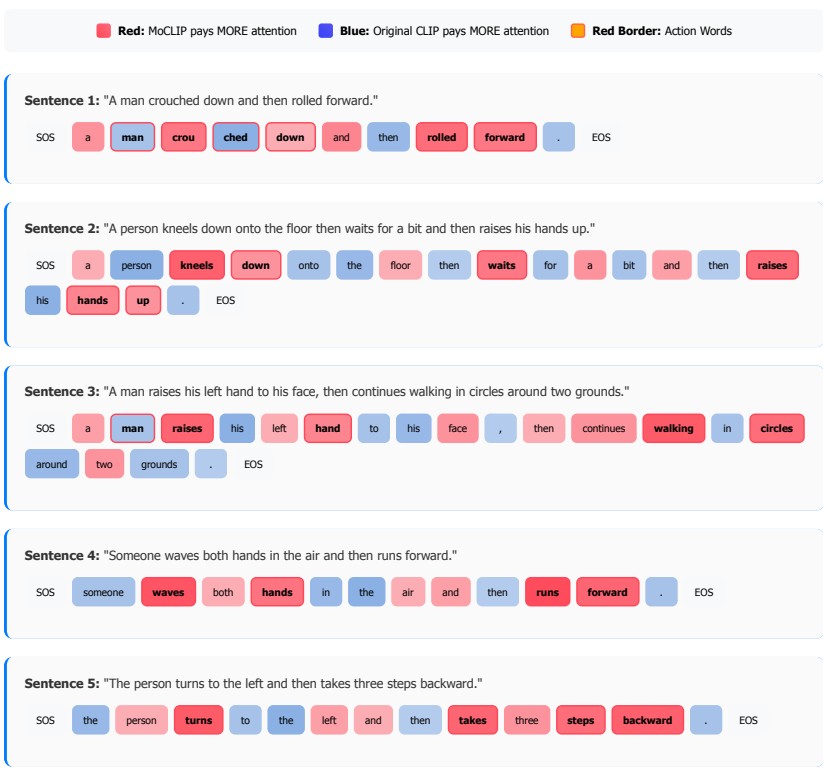

Figure 11: Comparison of final-layer attention between MoCLIP and original CLIP on motion descriptions. Red tokens receive more attention in MoCLIP, blue in original CLIP; red borders denote action words. MoCLIP consistently reallocates attention toward verbs and motion-centric tokens, highlighting its improved focus on the semantic core of actions.

## LLM USAGE STATEMENT

During the preparation of this manuscript, we strictly used a large language model (LLM) for language refinement purposes. Specifically, the LLM was used to improve the clarity, grammar, and readability of sentences originally written by the authors. The technical content, including the conceptual framework, methodology, mathematical formulations, experimental setup, and results, was entirely designed, developed, and validated by the authors without AI assistance.

The LLM was not used to generate novel ideas, algorithms, experimental designs, analyses, or citations. All scientific contributions and claims in this work originate solely from the authors. We also performed thorough reviews to ensure that the final text accurately reflects our intended meaning and that no fabricated references or unsupported claims were introduced during the editing process.

