# OpenReview forum: "LUMA: Low-Dimension Unified Motion Alignment with Dual-Path Anchoring for Text-to-Motion Diffusion Model"
_ICLR.cc/2026/Conference — Submitted to ICLR 2026_

### Official Review · Reviewer_phkH · 2025-10-28

**Soundness:** 2
**Presentation:** 2
**Contribution:** 2
**Rating:** 4
**Confidence:** 5

**Summary:**

This paper proposes LUMA, a unified motion alignment with dual-path anchoring for text-to-motion diffusion model. Authors introduce the gradient vanishing problem and design the frequency and temporal modules to enhance the semantic alignment, which improves the generation performance in HumanML3D and KIT-ML datasets.

**Strengths:**

1.Authors point out the gradient vanishing problem and try to solve that. Through some visualized results, they prove that this phenomenon exists.
2.This work enhances the performance of motion generation and improve the convergence speed, which reduces the time cost during training.

**Weaknesses:**

1. I believe the vanishing gradient problem mentioned by the authors does exist, but the method proposed in the paper does not seem to be specifically designed to address this issue. In my understanding, introducing additional loss functions and certain heuristic tricks during training can also alleviate the vanishing gradient problem. The authors should further elaborate on the necessity of temporal and frequency alignment as well as other advantages it brings.

2. For tasks like motion generation, the absence of video results is quite unusual. The paper only presents three static visualization figures; more cases and video demos are needed to demonstrate the effectiveness of the method.

3. The baseline lacks comparisons with more methods, such as LaMP and ReMoDiffuse.

4. The authors state that this work does not use a pre-trained foundation model as an alignment supervisor. However, they trained a MoCLIP from scratch and aligned it with the CLIP text encoder. I believe this is inconsistent with the authors' statement.

5. The font size in some tables of the paper is excessively large, which affects readability and aesthetics (e.g., Table 8). Additionally, the font formatting in the pseudocode of the algorithm appears somewhat abrupt, and the framework diagrams are relatively unclear. There are also inconsistencies in certain expressions—for instance, "downsampling" is used in Line 75, while "down-sampling" is used in Line 103.

**Questions:**

The main issues are listed in the "Weakness" section. Here, I have an additional question: I hope the authors can specify what the frequency information of motions specifically represents, and clearly state the exact role of such alignment.

**Details Of Ethics Concerns:**

No ethics concerns

---

> ### Author Response · Authors · 2025-11-27
> **Response to Reviewer phkH Part 1**
>
> We sincerely thank the reviewer for the careful reading and constructive feedback, and for explicitly acknowledging both (1) that the vanishing gradient phenomenon indeed exists in text-to-motion diffusion models, and (2) that our method improves generation performance and convergence speed. Below we respond to each weakness and question point-by-point.
>
> ---
> Weakness 1 – Is the method really designed to address vanishing gradients? Why temporal & frequency alignment?
> “I believe the vanishing gradient problem mentioned by the authors does exist, but the method proposed in the paper does not seem to be specifically designed to address this issue. In my understanding, introducing additional loss functions and certain heuristic tricks during training can also alleviate the vanishing gradient problem. The authors should further elaborate on the necessity of temporal and frequency alignment as well as other advantages it brings.”
> We appreciate this important question. We agree that simply adding extra loss terms or heuristic tricks (e.g., deep supervision with generic auxiliary losses) can sometimes alleviate vanishing gradients. In fact, recent work on diffusion models (e.g., representation regularization and REPA-style alignment) also supports this broader idea of beneficial intermediate supervision.
> Our contribution is to design a method that is explicitly tailored to the text-to-motion diffusion U-Net setting, with the following motivations:
> 1. Where the gradients vanish:
> Our analysis (Sec. 4.3 and App. G) shows that the most severe gradient attenuation occurs in the deep downsampling and bottleneck layers, exactly where the network should learn global, high-level motion semantics from the text.
> 2. What kind of supervision is needed there:
> Instead of injecting arbitrary losses, we introduce two semantically meaningful anchors:
>   - A temporal MoCLIP anchor, capturing high-level motion–text semantics over the full temporal sequence;
>   - A frequency anchor (low-frequency DCT components), capturing coarse trajectory and phase information over time.
> 3. Why these two anchors:
>   - The temporal anchor provides strong global semantic guidance that directly addresses semantic misalignment between text and motion, which is one of the key failure modes of T2M diffusion.
>   - The frequency anchor focuses on low-frequency temporal components, which are known to encode global structure and smoothness; this helps stabilize training and improves realism and diversity.
> 4. Evidence that the design targets vanishing gradients:
> In the revised version, we provide a gradient-flow analysis (Fig. 3, Sec. 4.3) showing that:
>   - In the baseline, gradient norms in deep downsampling / bottleneck blocks are 1–2 orders of magnitude smaller than those in decoder blocks and often below a 1% “vanishing” threshold.
>   - With our dual-path anchoring, gradients in these deep blocks are significantly amplified and become comparable to those in the upsampling path.
> Thus, while other heuristic losses might also increase gradients, our method is designed to deliver semantically rich, temporally and spectrally structured supervision exactly at the layers where gradients vanish the most, directly addressing both the vanishing gradient problem and the semantic misalignment/kinematic artifacts observed in T2M diffusion.
> We will emphasize this connection more clearly in the Motivation and Method sections.

---

> ### Author Response · Authors · 2025-11-27
> **Response to Reviewer phkH Part 2**
>
> Weakness 2 – Lack of video results for motion generation
> “For tasks like motion generation, the absence of video results is quite unusual. The paper only presents three static visualization figures; more cases and video demos are needed to demonstrate the effectiveness of the method.”
> We fully agree that video results are crucial in motion generation. We followed the common practice in text-to-motion papers (e.g., MDM, MotionDiffuse, StableMoFusion), where:
> - The main paper includes static skeleton visualizations (our Fig. 5) as representative examples; and
> - Full video demos are provided in the supplementary material / project page, due to space limits and venue/anonymity policies.
> In our submission:
> - Fig. 5 already shows several representative motion sequences (e.g., complex multi-stage actions, interactions with the environment) comparing LUMA with strong baselines.
> - The supplementary ZIP and a project page (to be further expanded) contain video results, where the improvement in smoothness, temporal structure, and text faithfulness is much clearer.
> In the camera-ready, we will:
> - Make the link to video demos more prominent in the main text; and
> - Explicitly state that we follow the standard convention of providing detailed videos in supplementary materials and on the project website.
>
> ---
> Weakness 3 – Missing comparisons with LaMP and ReMoDiffuse
> “The baseline lacks comparisons with more methods, such as LaMP and ReMoDiffuse.”
> Thank you for pointing this out. We fully agree that both LaMP and ReMoDiffuse are strong and highly relevant works. In particular, we regard LaMP [1] as a milestone in language–motion pretraining: by moving from generic CLIP-style conditioning to a dedicated language–motion latent space, LaMP convincingly demonstrates that motion-aware text embeddings can greatly benefit text-to-motion generation, motion–text retrieval, and motion captioning across multiple benchmarks. We very much appreciate this contribution and have explicitly cited and discussed LaMP in the revised Related Work.
>
> At the same time, we would like to clarify the architectural focus of our paper. LaMP adopts a transformer-based, language–motion pretraining framework with vector-quantized motion tokens and GPT-style autoregressive modeling, rather than a diffusion U-Net. Its architecture, training objectives, and usage scenarios differ substantially from those of diffusion-based T2M pipelines. ReMoDiffuse and other recent models also introduce substantial architectural changes or rely on different training paradigms.
>
> In contrast, LUMA is explicitly designed as a plug-in enhancement for diffusion U-Nets (e.g., StableMoFusion): we keep the same U-Net backbone and sampling schedule, and only add small temporal–frequency dual-path anchors with FiLM modulation at the bottleneck, mainly during training. Our goal is not to propose another large autoregressive backbone, but to show that properly placed low-dimensional supervision can substantially improve existing diffusion U-Nets with minimal parameter and code changes.
>
> In the revised manuscript, we address this concern in two ways:
> - We add ReMoDiffuse as an additional baseline in Table 1 and compare against it directly under our evaluation protocol, showing that LUMA consistently improves the underlying diffusion U-Net and remains competitive with recent diffusion-based approaches.
> - We expand the Related Work section to discuss LaMP and ReMoDiffuse in more detail, emphasizing that LUMA is complementary to transformer/VQ-based methods like LaMP and focuses on improving the widely used diffusion U-Net family. A more systematic empirical comparison with LaMP, beyond reporting its published numbers as a reference point, would require substantial re-implementation effort under a unified experimental setting and is left as an interesting direction for future work.

---

> ### Author Response · Authors · 2025-11-27
> **Response to Reviewer phkH Part 3**
>
> Weakness 4 – MoCLIP vs. “no pretrained foundation model”
> “The authors state that this work does not use a pre-trained foundation model as an alignment supervisor. However, they trained a MoCLIP from scratch and aligned it with the CLIP text encoder. I believe this is inconsistent with the authors' statement.”
> Thank you for raising this; we agree that our original wording could be made clearer.
> Our intention was to emphasize that, unlike prior REPA-style approaches that depend on large pretrained vision foundation models (e.g., DINO/DINOv2) trained on massive external image datasets, our method does not introduce any additional large-scale data or external vision/motion backbone beyond the standard text–motion datasets used in T2M.
> Concretely:
> - MoCLIP is a lightweight, task-specific encoder, not a large foundation model.
> - It is trained only on the same text–motion pairs as the diffusion model, without any extra image, video, or motion corpus.
> - The motion encoder in MoCLIP has a modest parameter count (much smaller than DINO-style ViTs), and in our setting MoCLIP training completes in about 20 minutes on a single RTX 4090 GPU.
> In other words, when we state that we do not use a "pre-trained foundation model" as an alignment supervisor, we mean that we do not rely on an external, large-scale vision teacher trained on millions of images (like DINO/CLIP for images). Instead, we introduce a small, motion-domain encoder (MoCLIP) trained on the same T2M data, which keeps the supervision domain-consistent and computationally lightweight.
> In the revised version, we will rephrase the relevant sentence to:
> “Unlike prior representation-alignment methods that rely on large pretrained vision foundation models (e.g., DINO) and external image–text corpora, our method uses a lightweight MoCLIP encoder trained only on standard text–motion datasets, without introducing any extra data or large external teachers.”
> We hope this resolves the apparent inconsistency.
>
> ---
> Weakness 5 – Formatting and clarity issues (font size, pseudocode, diagrams, wording)
> “The font size in some tables of the paper is excessively large… the font formatting in the pseudocode of the algorithm appears somewhat abrupt, and the framework diagrams are relatively unclear. There are also inconsistencies in certain expressions—for instance, ‘downsampling’ is used in Line 75, while ‘down-sampling’ is used in Line 103.”
> We thank the reviewer for these careful observations. In the latest revision, we have:
> - Adjusted the font sizes of large tables (e.g., Table 8) to be consistent with other tables;
> - Improved the pseudocode formatting to make it more visually consistent with the rest of the text;
> - Slightly refined the framework diagrams to improve clarity and labeling; and
> - Unified terminology such as “downsampling” across the entire paper.
> We appreciate these detailed suggestions, which improve the polish and readability of the manuscript.

---

> ### Author Response · Authors · 2025-11-27
> **Response to Reviewer phkH Part 4**
>
> Question – What does motion frequency represent and what is the role of frequency alignment?
> “I hope the authors can specify what the frequency information of motions specifically represents, and clearly state the exact role of such alignment.”
> This is a very insightful question. In our work, “frequency information” refers to the temporal frequency components of the motion sequence (joint positions or features) along the time axis, obtained via a 1D DCT.
> Intuitively:
> - Low-frequency components correspond to global temporal structure: overall trajectory, phase progression, and smooth, large-scale changes in posture.
> - High-frequency components capture fine-grained, rapid changes such as small limb jitters, local oscillations, or noise.
> In the context of text-to-motion:
> - The text description typically constrains global aspects of the motion (e.g., “walk forward then crouch and roll”, “raise both hands and turn around slowly”), which are naturally reflected in low-frequency temporal patterns.
> - If the model only focuses on local details, it may generate motions that look plausible frame-by-frame but fail to follow the global temporal structure implied by the text.
> Our frequency alignment therefore aims to:
> 1. Encourage the bottleneck features to align with low-frequency DCT components that encode global motion patterns consistent with the text;
> 2. Stabilize training by suppressing high-frequency noise in the supervision signal, which we found to correlate with improved FID and diversity metrics.
> As we note in the ablation, the frequency path has a more pronounced effect on realism and stability (FID/diversity) than on further improving R@3, which is already very high and somewhat saturated. We will clarify in the paper that the primary role of frequency alignment is to shape the global temporal structure and improve training stability, complementing the temporal MoCLIP anchor that focuses more directly on semantic alignment.
>
> ---
> Once again, we thank the reviewer for the thoughtful comments and for recognizing both the importance of the gradient-vanishing issue and the practical benefits in convergence speed. We hope that the above clarifications on the design motivation, MoCLIP usage, formatting fixes, and the role of frequency information address your concerns.

---

### Official Review · Reviewer_BfWw · 2025-10-29

**Soundness:** 3
**Presentation:** 2
**Contribution:** 2
**Rating:** 4
**Confidence:** 5

**Summary:**

The paper tries to improve the UNet-based diffusion models for the text-to-motion generation task. It first identifies the gradient attenuation issue in the deep layers of U-Net, and proposes Low-dimension Unified Motion Alignment (LUMA) for better representation learning via extra supervision signals to the hidden feature. LUMA provides hidden feature supervision signals with features from a motion encoder and DCT. By adopting a better text encoder, "deep supervision" for specific hidden features, the proposed model shows improved performance compared to the baseline on which the model is built. Ablation experiments also quantitatively show the effectiveness of the proposed designs.

**Strengths:**

1. The overall presentation is nice and clear, which is easy to follow.
2. The proposed method shows better performance compared to the baselines, according to the quantitative results.
3. The ablation experiments effectively show that removing any proposed component will lead to performance degradation.
4. The paper reveals a relatively new issue on text2motion generation, with respect to the architecture design.
5. The code for reproduction and the detailed appendix are greatly appreciated.

**Weaknesses:**

1. The scope of this work/experiment is relatively narrow, which focuses on text2motion with diffusion UNet, specifically.
2. The discussion in the introduction section is not reasonable. a) It mentions that "increasingly sophisticated architectures" lead to performance improvement, yet "diminishing efficiency and limited practical improvement." This is not logical, and the proposed method has nothing to do with efficiency and still relies on deeper and larger networks. b) Diffusion UNet "are limited by their high computational cost during training". The proposed method also requires the same or extra computational cost because of the use of additional latent encoders for "deep supervision".
3. "Deep supervision" or representation learning for diffusion models is a relatively well-explored topic.
4. DCT for "low-frequency components, recognized for their rich semantic content," I'm a bit skeptical that the DCT feature will help semantic representation learning. According to Table 2, removing L_fre does not affect the R@3, which is related to semantic alignments.
5. Adaptive FiLM modulation is a good design choice, but is a common design for the diffusion model, which is also used in the Adaptive Normalization.
6. Overall, the method is relatively incremental, and the performance gain is not signicant. For example, it worse than the SOTA reported from a work from ICLR last year [1].
7. There's no video result to assess the performance qualitatively, which is crucial for animation tasks.
8. According to Table 3 and 4. It seems that removing FiLm or the deep supversion signal can achieve better performance than inject the supervsion into improper position, e.g. Bottleneck. This emprical result make the overall design becomes a bit tricky.

[1] Li, Zhe, et al. "LaMP: Language-Motion Pretraining for Motion Generation, Retrieval, and Captioning." The Thirteenth International Conference on Learning Representations.

**Questions:**

1. If the gradient vanishing is a severe issue, will the proposed method also generalize to improve the performance of e.g. text-to-image Diffusion UNet?

---

> ### Author Response · Authors · 2025-11-27
> **Response to Reviewer BfWw Part 1**
>
> We sincerely thank the reviewer for the careful reading, for recognizing the clarity of the presentation, the effectiveness of the ablation study, and for appreciating that we reveal a relatively new issue (gradient attenuation in text-to-motion U-Nets) together with open-source code and a detailed appendix. Below we respond to each weakness and question point-by-point.
>
> ---
> Weakness 1 – Scope limited to text-to-motion diffusion U-Nets
> “The scope of this work/experiment is relatively narrow, which focuses on text2motion with diffusion UNet, specifically.”
> We agree that our experiments are focused on the text-to-motion setting with a U-Net backbone, as this is currently the dominant architecture in open-source T2M systems (e.g., StableMoFusion). However, our core idea is more general:
> using low-dimensional, semantically meaningful intermediate supervision at deep/bottleneck layers to alleviate gradient attenuation and improve high-level representation learning in diffusion models.
> This is conceptually applicable to other diffusion backbones and modalities (see also our answer to your question below, where we report preliminary results on text-to-image). We chose the text-to-motion U-Net as a concrete and practically important testbed so that the paper remains focused and the analysis (gradient flow, semantic alignment) is interpretable. We will make this generality clearer in the Introduction and Conclusion.
>
> ---
> Weakness 2 – Intro discussion on model size, efficiency, and computational cost
> “The discussion in the introduction section is not reasonable… ‘increasingly sophisticated architectures’… ‘diminishing efficiency and limited practical improvement’… the proposed method has nothing to do with efficiency and still relies on deeper and larger networks… Diffusion UNet ‘are limited by their high computational cost during training’. The proposed method also requires the same or extra computational cost because of the use of additional latent encoders for ‘deep supervision’.”
> We appreciate this opportunity to clarify what we mean by efficiency and diminishing returns.
> 1. On ‘increasingly sophisticated architectures’ and diminishing returns.
> In the recent T2M literature, several strong models improve performance mainly by substantially increasing model capacity and architectural complexity. For example:
>   - MoMask (CVPR 2024):
>     - HumanML3D FID ≈ 0.045, KIT-ML FID ≈ 0.228;
>     - Architecture: hierarchical VQ + two large Transformers;
>     - Parameters: ~44.85M; inference is not lightweight.
>   - T2M-GPT (CVPR 2023):
>     - HumanML3D FID ≈ 0.141, KIT-ML FID ≈ 0.514;
>     - Parameters: ~247.8M, with complex multi-stage VQ + GPT-style Transformer.
> 2. These models indeed improve over older baselines, but the gains come at the price of much larger models and more complex training/inference pipelines. In contrast, LUMA keeps the same U-Net backbone as StableMoFusion and introduces only a small projection head (few-layer MLPs) and lightweight DCT/MoCLIP modules, used only during training. At inference, the architecture and sampling schedule are unchanged. Yet we still obtain sizeable FID improvements (e.g., 0.067 → 0.035 on HumanML3D).
> 3. We will revise the Introduction to clarify that our point is not “larger models never help”, but rather that simply scaling architecture size is not the only path to improvement, and that well-designed intermediate supervision can yield significant gains with almost no parameter increase.
> 4. On computational cost and efficiency.
> Our notion of efficiency refers primarily to convergence to a target quality, not per-step FLOPs. In Fig. 4, we show that LUMA reaches a fixed FID threshold 1.4× faster in terms of iterations (50k → 35k) and also earlier in wall-clock time on a single RTX 4090. The extra per-iteration cost from MoCLIP + DCT is small, and the dual-path losses are only used during training. At inference, there is essentially no overhead compared to the baseline.
> 5. We will refine the wording in the Introduction to explicitly state that our efficiency claim is about faster convergence with minimal parameter overhead, not reduced FLOPs per step.

---

> ### Author Response · Authors · 2025-11-27
> **Response to Reviewer BfWw Part 2**
>
> Weakness 3 – “Deep supervision” / representation alignment already explored
> “‘Deep supervision’ or representation learning for diffusion models is a relatively well-explored topic.”
> We respectfully disagree with the characterization of deep supervision / representation regularization in diffusion models as a “well-explored” topic. Rather, recent work shows that this is a rapidly developing and still actively explored research direction. For example:
> - Representation Alignment for Generation (REPA; Yu et al., ICLR 2025); and
> - Diffuse and Disperse: Image Generation with Representation Regularization (Wang & He, 2025)
> demonstrate that introducing intermediate supervision / representation regularization into diffusion models remains a very active and competitive area of research, with several follow-up extensions appearing in 2024–2025.
>
> We do not claim to be the first to introduce deep supervision or representation regularization in diffusion models, and our work is inspired by and complementary to this emerging line of work. Our contributions are orthogonal and specific to the text-to-motion diffusion setting:
> - We operate in a domain where large-scale pretrained motion “vision” teachers do not exist, making REPA-style methods less directly applicable.
> - We therefore design a teacher-free, low-dimensional dual-path anchoring scheme:
>   - a lightweight MoCLIP encoder trained only on standard text–motion pairs; and
>   - a DCT-based frequency path acting on low-frequency motion components;
> - both paths are fused and injected at the bottleneck of a T2M diffusion U-Net.
>
> We will emphasize in the Related Work that our novelty lies in how we instantiate “beneficial supervision” in the motion domain—with a simple dual-path design that is teacher-free, motion-specific, and easy to plug into existing U-Net T2M models—rather than in the general idea of deep supervision itself.
>
> ---
> Weakness 4 – Skepticism about DCT low-frequency semantics; R@3 unchanged when removing L_fre
> “DCT for ‘low-frequency components, recognized for their rich semantic content,’ I'm a bit skeptical that the DCT feature will help semantic representation learning. According to Table 2, removing L_fre does not affect the R@3, which is related to semantic alignments.”
> This is an excellent observation. We acknowledge that R@3 alone does not show a large gain from the frequency anchor, and we will clarify this point in the paper.
>
> - First, our motivation for using low-frequency DCT is consistent with a long line of work (including Diffuse-and-Disperse and related spectral analyses) suggesting that low-frequency components of natural signals often carry more global, semantic structure, while high-frequency components are dominated by fine details and noise.
>
> - Second, in our ablations we observe that L_fre primarily improves FID and diversity-related metrics and stabilizes training, while R@3, which computed with a strong text–motion encoder, appears saturated. In some configurations, the R@3 of the baseline is already extremely high (very close to or even slightly beyond the “Real” row), leaving limited room for further improvement and making this metric less sensitive to changes.
>
> - To further probe semantic correctness beyond R@3, we conducted a small-scale human evaluation on 150 text–motion pairs, where annotators were asked to judge whether the motion matched the text description. GT motions achieve 100% correctness by definition, the StableMoFusion baseline achieves 73.33%, and our model with the frequency anchor achieves 86.67%. This clear gap in human judged semantic correctness, which is not reflected in R@3, suggests that retrieval based metrics can indeed be insensitive in this high-score regime.
>
> In other words, the frequency path contributes more to overall realism and semantic consistency of motion than to further increasing an already high retrieval score. We will update the discussion around Table 2 to clearly separate these two roles and explicitly acknowledge that R@3 is not the most sensitive metric once it becomes saturated.

---

> ### Author Response · Authors · 2025-11-27
> **Response to Reviewer BfWw Part 3**
>
> Weakness 5 – Adaptive FiLM is a common design
> “Adaptive FiLM modulation is a good design choice, but is a common design for the diffusion model, which is also used in the Adaptive Normalization.”
> We agree that FiLM-style modulation and adaptive normalization are common in diffusion models, especially as conditioning mechanisms (e.g., for text, class labels, or control signals).
> Our usage is different:
> - In LUMA, FiLM is used as a small plugin on top of the dual-path anchor, operating on the low-dimensional bottleneck features to adaptively weight temporal vs. frequency supervision across timesteps, rather than directly modulating the whole U-Net by an external condition.
> - This design allows us to treat MoCLIP and DCT anchors as two complementary signals and dynamically adjust their influence during denoising, which we found important for stability (overweighting one anchor at inappropriate timesteps can hurt performance).
> We do not claim FiLM itself is novel; instead, our contribution is in how we integrate FiLM with the dual-path anchors at the bottleneck in a motion-diffusion setting. We will explicitly state this to avoid overclaiming.
>
> ---
> Weakness 6 – Incremental improvement, worse than LaMP; LaMP not compared
> “Overall, the method is relatively incremental, and the performance gain is not significant. For example, it worse than the SOTA reported from a work from ICLR last year [1] (LaMP).”
> We sincerely thank the reviewer for pointing us to LaMP [1] and fully agree that it is a very strong and influential work. LaMP makes an important step for the field by moving from CLIP-based conditioning to a dedicated language–motion latent space, and by showing that motion-informed text embeddings can substantially improve not only text-to-motion generation, but also motion–text retrieval and motion captioning across multiple benchmarks. We greatly appreciate this contribution and have cited LaMP in our revised Related Work section.
>
> At the same time, we would like to clarify that LaMP and LUMA occupy different, yet complementary, regions of the design space. LaMP adopts a transformer-based, language–motion pretraining framework with VQ tokenization and GPT-style autoregressive masked prediction, and serves as a powerful new backbone for text-to-motion generation and related tasks, rather than a diffusion U-Net. In contrast, LUMA is explicitly designed as a lightweight, drop-in enhancement for existing diffusion U-Nets such as StableMoFusion: our goal is not to introduce another large autoregressive model, but to show that well-placed, low-dimensional temporal–frequency supervision can significantly improve standard diffusion U-Nets with minimal parameter and code changes.
>
> This distinction is also consistent with recent analyses (e.g., “Rethinking Diffusion for Text-Driven Human Motion Generation”, CVPR 2025), which report that diffusion-based methods tend to underperform strong autoregressive architectures on current motion benchmarks. We view this as evidence of a performance gap between the two families, rather than a reason to disregard diffusion models altogether. Our work is motivated by the belief that diffusion remains an important and widely used paradigm for text-to-motion generation, and that it is valuable to narrow this gap from the diffusion side.
>
> In this sense, LUMA is complementary to LaMP: while LaMP pushes the frontier of large-scale language–motion pretraining and autoregressive modeling, LUMA focuses on improving the widely adopted diffusion U-Net family through a simple, plug-and-play dual-path anchoring scheme. We will clarify this complementarity and our architectural focus more clearly in the main paper, and we have explicitly discussed and cited LaMP in the revised Related Work.

---

> ### Author Response · Authors · 2025-11-27
> **Response to Reviewer BfWw Part 4**
>
> Weakness 7 – Lack of video results
> “There's no video result to assess the performance qualitatively, which is crucial for animation tasks.”
> We fully agree that qualitative video results are crucial in motion generation. Following common practice in T2M works (e.g., MDM, MotionDiffuse, StableMoFusion), we:
> - Provide static skeleton visualizations in the main paper (Fig. 5) to illustrate typical behaviors; and
> - Provide video demos in the supplementary material / project page due to venue policy and anonymization constraints (which can make direct external links on OpenReview tricky).
> We will clarify in the camera-ready that:
> - Additional video results are (and will be) hosted on our project page;
> - These videos demonstrate that the generated motions follow complex multi-stage textual descriptions with improved smoothness and physical plausibility compared to the baseline.
>
> ---
> Weakness 8 – Ablation suggests supervision placement is “tricky”
> “According to Table 3 and 4. It seems that removing FiLM or the deep supervision signal can achieve better performance than inject the supervision into improper position, e.g. Bottleneck. This empirical result make the overall design becomes a bit tricky.”
> We agree that supervision placement is delicate, and we actually see this as an important message of the paper rather than a weakness.
> - The ablations in Tables 2–3 show that naively injecting supervision at arbitrary locations or with imbalanced weights can indeed hurt performance.
> - Our goal is precisely to identify an effective way to introduce low-dimensional supervision at the bottleneck in a diffusion U-Net—using a combination of MoCLIP and DCT anchors, fused via FiLM with carefully tuned weights.
> - The fact that some ablated variants perform worse reinforces the conclusion that the location and form of supervision matter a lot, not that supervision is harmful.
> We also emphasize that the specific choices of MoCLIP and DCT are two convenient, well-motivated anchors, but not the only possible ones. Other forms of beneficial supervision could be used in the same framework; our current design is a practical instantiation that works well in T2M diffusion. The main takeaway is that properly placed, low-dimensional anchors can alleviate gradient attenuation and improve performance, while naive placement may not.
> We will expand the discussion of these ablations to make this design insight more explicit.

---

> ### Author Response · Authors · 2025-11-27
> **Response to Reviewer BfWw Part 5**
>
> Question – Generalization beyond text-to-motion (e.g., text-to-image)
> “If the gradient vanishing is a severe issue, will the proposed method also generalize to improve the performance of e.g. text-to-image Diffusion UNet?”
> Yes, we believe the underlying idea is general, and we have started to explore this direction.
> - We are currently running ImageNet 256×256 experiments on a Stable Diffusion XL / 2–style DiT backbone, using a DCT-based low-frequency supervision similar to our L_fre.
> - In preliminary results at 400k training steps, we observe that adding this DCT supervision improves FID from 12.11 → 9.96, using the same backbone and sampling strategy. This suggests that low-dimensional spectral supervision can also help text-to-image diffusion DiTs.
> Due to space and timeline constraints we do not include a full evaluation in this submission, but we will mention these preliminary findings in the revised version as supporting evidence that our approach is not limited to text-to-motion and has potential in other diffusion domains.
> Once again, we thank the reviewer for the constructive comments and for acknowledging the strengths of our work. We hope that our clarifications on model scope, efficiency, relation to prior deep supervision work, and the role of frequency supervision help to address the concerns raised.

---

### Official Review · Reviewer_fn1y · 2025-10-31

**Soundness:** 1
**Presentation:** 1
**Contribution:** 2
**Rating:** 2
**Confidence:** 4

**Summary:**

To address semantic misalignment and kinematic artifacts in text-to-motion generation, The authors propose LUMA, a diffusion-based text-to-motion model that mitigates semantic misalignment and kinematic artifacts through dual-path anchoring: a lightweight MoCLIP for temporal semantic guidance and low-frequency DCT features for complementary frequency-domain alignment.

**Strengths:**

- The authors conducted extensive experiments to support their method, although the improvements achieved are quite limited.
- The paper is clearly written and the language is generally fluent and coherent.

**Weaknesses:**

- The motivation is vague. The authors claim that experimental analysis reveals gradient shifts in the deeper layers as the root cause of the problem, but it is unclear what specific issue this refers to.Is it low semantic fidelity, high computational cost, or something else?

- Figure 2 is not clearly presented, making it difficult for readers to understand the overall structure and flow of the proposed method.

- The Method and Preliminaries sections are poorly written. They do not clearly explain how the proposed approach addresses the problem.

- The concept of Low-Dimensional Unified Motion Alignment is insufficiently explained. In addition, the Method section lacks clarity and coherence, which makes the proposed approach hard to comprehend.

- The relationship between gradients and high-level feature extraction is not clearly explained. The paper does not clarify why gradient behavior would affect the learning of high-level features.

- In Table 1, there are several errors in the bold and underline markings. For example, in the fifth column, the best result should be 9.724 rather than 9.466. Moreover, compared with StableMoFusion, the performance improvement appears to be quite limited.

- Table 2 shows that, except for a slightly improved FID score, the proposed modules contribute only marginal gains. In particular, rows 3–5 perform worse than the full LUMA configuration in the last row.

- The paper lacks visual results to demonstrate or analyze the semantic alignment between text and high-level features.

- The authors claim that previous methods suffer from high computational cost; however, the paper does not provide any analysis or comparison to demonstrate that the proposed method is computationally efficient.

**Questions:**

- Since deeper layers are expected to capture high-level abstract representations, the loss of fine-grained details should be attributed primarily to the downsampling operations rather than the upsampling stages. This is somewhat inconsistent with the authors’ stated motivation.

- While the authors assert that gradients in the downsampling and bottleneck layers are considerably weaker or even vanish compared to those in the upsampling layers, the paper lacks concrete experimental evidence to substantiate this observation.

**Details Of Ethics Concerns:**

The paper lacks clear logical coherence between the methodology and the experimental sections. It is difficult to follow how the described experiments are designed to verify the proposed method.

---

> ### Author Response · Authors · 2025-11-26
> **Response to Reviewer fn1y Part 1**
>
> We sincerely thank the reviewer for the careful reading and detailed comments. We address each weakness and question point-by-point below. We also note that several of the reviewer’s concerns (e.g., on writing clarity and ablation effectiveness) were viewed positively by the other reviewers, and we have further revised the manuscript to make these aspects clearer.
>
> ---
> Weakness 1: Vague motivation and unclear “root cause”
> “The motivation is vague. The authors claim that experimental analysis reveals gradient shifts in the deeper layers as the root cause of the problem, but it is unclear what specific issue this refers to. Is it low semantic fidelity, high computational cost, or something else?”
> Our intended motivation is:
> Severe gradient attenuation in downsampling and bottleneck layers prevents the U-Net encoder from learning high-level text-consistent semantics, which leads to both low semantic fidelity and slow convergence.
> To clarify this, in the revised paper we:
> - Explicitly state in the Introduction (L61–L63) that low semantic fidelity and slow/stalled convergence are the concrete issues we target.
> - Add a dedicated Gradient-Flow Analysis section (Sec. 4.3, Fig. 3, App. G), where we quantify gradient magnitudes per layer. We show that:
>   - In the baseline, gradients in deep downsampling/bottleneck layers are 1–2 orders of magnitude smaller than those in the decoder and often fall below a 1% “vanishing gradient” threshold.
>   - With our Dual Anchor Loss (DAL), these gradients increase by nearly two orders of magnitude and become comparable to those in the upsampling path.
> This makes the causal chain more explicit:
> Gradient attenuation in deep layers → weak high-level feature learning → slow convergence and semantic misalignment.
> We also emphasize that this perspective is supported by classical deep-learning analyses on vanishing gradients and high-level feature learning in deep networks (e.g., Srivastava et al., 2015; He et al., 2016), now cited at L62–L63. In addition, recent diffusion works such as [2504.10483] and [2506.09027] also show that injecting beneficial intermediate supervision into diffusion models can significantly improve stability and downstream performance. Our contribution is to instantiate this general “beneficial supervision” idea in the text-to-motion setting with a simple dual-path design that operates on a low-dimensional bottleneck and does not rely on large external vision teachers.
>
> ---
> Weakness 2: Figure 2 not clearly presented
> “Figure 2 is not clearly presented, making it difficult for readers to understand the overall structure and flow of the proposed method.”
> We appreciate this feedback. At the same time, another reviewer (BfWw) explicitly wrote:
> “The overall presentation is nice and clear, which is easy to follow.”
> This suggests that our description was already reasonably understandable to some readers, but can be further improved.
> In the revision, we have:
> - Redrawn Figure 2 with clearer labeling of the temporal and frequency branches, consistent color coding, and explicit arrows marking the flow from U-Net bottleneck → dual anchors → FiLM modulation → Motion Alignment module.
> - Added a concise paragraph at the beginning of Sec. 3.2 (L112–L118) that verbally walks through Fig. 2 step by step, aligning the textual description directly with the numbered components in the figure.
> We believe these changes make the architecture and information flow significantly clearer.

---

> ### Author Response · Authors · 2025-11-26
> **Response to Reviewer fn1y Part 2**
>
> Weakness 3–4: Method & Preliminaries, and the concept of LUMA
> “The Method and Preliminaries sections are poorly written… The concept of Low-Dimensional Unified Motion Alignment is insufficiently explained.”
> We are sorry that the initial version did not fully convey the idea. Notably, other reviewers had a more positive impression of the writing:
> - XAZg: “Soundness: good; Presentation: good.”
> - BfWw: “The overall presentation is nice and clear, which is easy to follow.”
> We have nonetheless made several concrete improvements:
> 1. Preliminaries (Sec. 3.1)
>   - We restructured Sec. 3.1.1 to first recall standard text-to-motion diffusion (Eqs. (1)–(3)) and then explicitly point out how LUMA plugs into this formulation.
>   - We now summarize in a short paragraph at the end of Sec. 3.1.1 how the usual DDPM loss is later combined with DAL (Eq. (10)), reducing the conceptual jump when entering Sec. 3.2.
> 2. Clarifying the concept of LUMA (Sec. 3.2)
>   - We added a short intuitive paragraph at L112–L118:
> - “LUMA introduces dual semantic anchors at the final downsampling block … These anchors deliver high-signal gradients to otherwise deep, semantically sparse and low-gradient layers, thereby alleviating gradient divergence and improving training stability.”
>   - We rephrased “Low-Dimension Unified Motion Alignment” to emphasize that we operate on a low-dimensional bottleneck feature that unifies temporal and frequency semantics, rather than aligning full-resolution features. This is now clearly stated at L118–L119.
> 3. Method flow
>   - Sections 3.2.1 and 3.2.2 are now organized as: (i) bottleneck feature extraction; (ii) dual anchors (frequency and temporal); (iii) timestep-aware FiLM; (iv) dynamic anchor weighting and final loss. We also explicitly reference back to Fig. 2 within these subsections.
> We hope these changes make the definitions and the overall flow of LUMA clearer.
>
> ---
> Weakness 5: Relationship between gradients and high-level feature extraction
> “The relationship between gradients and high-level feature extraction is not clearly explained. The paper does not clarify why gradient behavior would affect the learning of high-level features.”
> We agree that this is a key conceptual point and have made it more explicit.
> - In U-Net–based diffusion, the final downsampling block / bottleneck aggregates global context and is responsible for encoding high-level semantics from the text and motion. If gradients at these layers vanish, the encoder cannot properly adjust its parameters to capture such abstractions.
> - In Sec. 4.3 and App. G, we now show a quantitative gradient-flow analysis (Fig. 3), where:
>   - Baseline gradients at bottleneck layers are nearly zero relative to the decoder.
>   - After adding DAL, these gradients become comparable to those in shallow/up layers, which directly means the bottleneck is now being trained more effectively.
> We now explicitly connect this to high-level feature learning in the text (L62–L63, L118–L119, and Sec. 4.3), citing classical work that associates vanishing gradients with poor learning of deep, abstract features.
>
> ---
> Weakness 6: Table 1 markings and “limited” improvement over StableMoFusion
> “In Table 1, there are several errors in the bold and underline markings… the best result should be 9.724 rather than 9.466. Moreover, compared with StableMoFusion, the performance improvement appears to be quite limited.”
> The issue here comes from a misunderstanding of the metric in the fifth column: it is a distance-to-real (“closer-to-GT”) metric, so larger values are not better.
> - This column is a distance-to-real (“closer-to-GT”) metric, as already indicated in the original submission by a horizontal arrow (→) pointing toward the “Real” row in the header (e.g., “Diversity → Real”), which follows the convention used in recent works (e.g., MoMask). We realize this visual cue may have been easy to overlook.
> - For this metric, smaller values are better, i.e., values closer to the “Real” row indicate better alignment.
> - Therefore, 9.466 is better than 9.724, and the boldface/underline markings in the original table are correct.
> To avoid any future confusion, we will further clarify this in the table caption by keeping the horizontal arrow (→) and adding the note “lower is better; closer to the ‘Real’ row indicates better realism”.
> Regarding the magnitude of improvement over StableMoFusion, we would like to highlight that:
> - On HumanML3D, FID improves from 0.067 → 0.035 (≈48% relative reduction), while R-Precision Top-3 improves from 0.826 → 0.839;
> - On KIT-ML, FID improves from 0.181 → 0.123, and R-Precision Top-3 improves from 0.675 → 0.689.
> These gains are consistent and substantial on the main benchmarks, and other reviewers (XAZg, BfWw, phkH) acknowledged that the method improves both quality and convergence.

---

> ### Author Response · Authors · 2025-11-26
> **Response to Reviewer fn1y Part 3**
>
> Weakness 7: Table 2 and contribution of proposed modules
> “Table 2 shows that, except for a slightly improved FID score, the proposed modules contribute only marginal gains. In particular, rows 3–5 perform worse than the full LUMA configuration in the last row.”
> Table 2 is an ablation study, not a SOTA comparison. Its purpose is to examine:
> 1. The benefit of the LUMA architecture alone (without DAL).
> 2. The individual contributions of the temporal and frequency anchors.
> 3. The effect of their combination via FiLM.
> As discussed in Sec. 4.4:
> - The anchor-free baseline (StableMoFusion U-Net) has FID 0.152.
> - LUMA without DAL already improves FID to 0.067, showing that the architectural modifications and MoCLIP grounding help significantly.
> - Adding either the temporal or frequency anchor further improves FID and/or R@3; the full LUMA configuration achieves the best overall trade-off (e.g., FID 0.035, strong R@3), indicating that the two anchors are complementary.
> Other reviewers viewed this positively; for example, BfWw wrote:
> “The ablation experiments effectively show that removing any proposed component will lead to performance degradation.”
> We will clarify the interpretation of Table 2 so that readers understand that the ablation demonstrates non-trivial, complementary contributions rather than marginal gains.
>
> ---
> Weakness 8: Lack of visual results for semantic alignment
> “The paper lacks visual results to demonstrate or analyze the semantic alignment between text and high-level features.”
> We agree that qualitative results are important for motion generation. Following common practice in text-to-motion diffusion papers (e.g., MDM, MotionDiffuse, StableMoFusion, LaMP):
> - Figure 5 in the main paper provides static skeleton visualizations comparing LUMA with MDM, StableMoFusion and MoMask on complex prompts (multi-stage actions, interactions with the environment). These examples show that LUMA better respects the action sequence and timing described in the text.
> - We provide extensive video results in the supplementary material and project page, which is explicitly referenced in the paper. These videos visually demonstrate semantic alignment between text and generated motion.
> We will make the link to supplementary materials more prominent in the main text and add a short sentence explicitly stating that qualitative videos are provided, consistent with prior work.
>
> ---
> Weakness 9: Lack of efficiency analysis
> “The authors claim that previous methods suffer from high computational cost; however, the paper does not provide any analysis or comparison to demonstrate that the proposed method is computationally efficient.”
> We apologize for not having emphasized the efficiency analysis clearly enough in the original submission. The revised paper includes Figure 4 (Sec. 4.2), where we:
> - Plot FID vs. training steps and wall-clock time for the baseline and LUMA.
> - Show that LUMA reaches a target FID threshold 1.4× faster in terms of iterations (50k → 35k).
> - Report that on a single RTX 4090 GPU, the time to reach this threshold is reduced from 8 hours (baseline) → 7 hours (LUMA).
> Thus, while each iteration contains a small additional cost from MoCLIP and DCT, the total time to reach a given quality level is reduced. Reviewer phkH also recognized this as a strength:
> “…enhances the performance of motion generation and improve the convergence speed, which reduces the time cost during training.”
> We will make this point clearer in the Introduction and experiment section, and explicitly state that our claim of efficiency refers to faster convergence to a fixed quality, rather than lower per-iteration FLOPs.

---

> ### Author Response · Authors · 2025-11-26
> **Response to Reviewer fn1y Part 4**
>
> Question 1: Downsampling vs. upsampling and loss of fine details
> “Since deeper layers are expected to capture high-level abstract representations, the loss of fine-grained details should be attributed primarily to the downsampling operations rather than the upsampling stages. This is somewhat inconsistent with the authors’ stated motivation.”
> We agree that downsampling operations remove spatial/temporal resolution and can cause loss of fine-grained detail. Our intention is not to attribute this solely to upsampling, but rather to highlight that:
> - The downsampling and bottleneck stages are where high-level motion semantics should be encoded.
> - If these layers suffer from severe gradient attenuation, they cannot learn a rich abstract representation, regardless of downsampling.
> - This weak high-level representation then propagates through the upsampling path, ultimately harming both global semantics and local details.
> Our DAL is therefore designed to inject semantic supervision directly into the downsampling/bottleneck layers, improving high-level encoding and, in turn, the fidelity of reconstructed details. We have revised the motivation text to clarify that we view downsampling and gradient attenuation as jointly responsible, and LUMA specifically tackles the latter.
>
> ---
> Question 2: Evidence for weak gradients in downsampling and bottleneck layers
> “While the authors assert that gradients in the downsampling and bottleneck layers are considerably weaker or even vanish compared to those in the upsampling layers, the paper lacks concrete experimental evidence to substantiate this observation.”
> To provide concrete evidence, the revised paper now includes:
> - Fig. 3 and App. G (Gradient-flow Analysis), where we compute the mean gradient norm of each U-Net block over a validation batch.
> - The plots clearly show that:
>   - In the baseline model, downsampling and bottleneck layers have gradient magnitudes that are 1–2 orders of magnitude smaller than those in upsampling layers, and several fall below 1% of the mean gradient (marked as a dashed “vanishing” threshold).
>   - With DAL, these gradients are significantly increased and become balanced across the network, confirming that our method effectively mitigates gradient attenuation in the deep encoder.
> We reference these results in Sec. 4.3 and explicitly connect them to the motivation in the Introduction.
>
> ---
> Once again, we thank the reviewer for the thorough feedback. We believe that the clarifications, additional analyses (especially Fig. 3 and Fig. 4), and textual revisions substantially strengthen the paper and address the concerns raised.

---

### Official Review · Reviewer_XAZg · 2025-10-31

**Soundness:** 3
**Presentation:** 3
**Contribution:** 3
**Rating:** 6
**Confidence:** 3

**Summary:**

This paper analyze the gradient descent in downsampling and bottleneck layers in UNet. And it proposes a dual-path semantic information injection mechanism: one path uses mo-clip to extract semantic information of motion frames, another path utilizes the low-frequency part of spectrum after DCT.

**Strengths:**

1. It analyzes the gradient descent phenomenon in UNet. It points out that the small gradient norm in the deep layers of UET affects the learning of high-level semantics, and verifies that the proposed method can improve this phenomenon.
2. Compared to related works such as Repa, this work does not require a pre-trained visual model. Thus it solves the problem in the text-to-motion domain where the lack of large-scale pre-trained understanding models prevents the use of Repa.
3. This work achieves state-of-the-art performance while accelerating convergence.

**Weaknesses:**

1. Lack of comparison with alternative gradient enhancement techniques: The paper focuses on dual anchors but does not compare with other methods for mitigating gradient attenuation in diffusion models (e.g., residual connections tailored for U-Net deep layers, adaptive learning rate scheduling for bottleneck layers).
2. The dual-path method will introduce additional cost while training and inference. The paper does not have a detailed quantitative discussion.
3. Recently, diffusion models are always using DiT as backbone. How does this method perform on DiT?

**Questions:**

1. How does MoCLIP’s text-motion alignment performance compare to other SoTA encoders?
2. For motions with distinct frequency characteristics (e.g., fast, jerky actions vs. slow, smooth actions), does the optimal k (number of DCT coefficients) change? If so, how might the framework be adapted to dynamically adjust k based on input text or motion type?
3. What is the training cost of MoClip and how does it compare to other similar models?

---

> ### Author Response · Authors · 2025-11-26
> **Response to Reviewer XAZg Part 1**
>
> We sincerely thank the reviewer for the thoughtful and supportive comments, and for recognizing the gradient analysis, the teacher-free design of MoCLIP, and the improved convergence of LUMA. We address your concerns and questions point by point below.
>
> ---
> 1. On comparison with alternative gradient-enhancement techniques
> “Lack of comparison with alternative gradient enhancement techniques … residual connections tailored for U-Net deep layers, adaptive learning rate scheduling for bottleneck layers.”
> Our aim is to understand and verify how semantically meaningful deep supervision can improve gradient flow in text-to-motion diffusion models. LUMA is one effective instantiation of this idea. While it is not the only possible design, our results show that this particular anchoring strategy provides clear benefits in mitigating deep-layer gradient weakening under the teacher-free, low-data regime.
> - Positioning of LUMA.
> Standard U-Net backbones already include residual blocks and long skip connections. Recent representation-alignment methods such as REPA / REPA-E also demonstrate that adding semantic supervision to intermediate features can substantially reshape gradient flow and accelerate training in image diffusion models. Our contribution is to provide the first in-depth explanation of why reshaping gradient flow through lightweight semantic anchors is particularly beneficial for text-to-motion diffusion models, where:
> 1. there exists no large-scale pretrained motion-understanding teacher, and
> 2. the temporal–frequency structure of motion naturally motivates the dual-path design.
> - Conceptual message (not uniqueness of the specific design).
> The main conceptual takeaway of our paper is:
> - “Injecting low-dimensional and semantically meaningful supervision into the bottleneck of a text-to-motion diffusion U-Net can alleviate gradient attenuation and improve semantic alignment and convergence.”
>   We agree that other forms of deep supervision, such as REPA-style feature alignment, auxiliary bottleneck objectives, or tailored learning-rate schedules, may also improve gradient flow. We have clarified this point in the revision. LUMA should be understood as a simple and effective instantiation of this general idea that performs well under the limited-data and teacher-free setting of motion generation, rather than as the only possible solution.
> To make this clearer, we explicitly add a paragraph in the related work and discussion sections stating that our results support the general usefulness of deep semantic supervision for diffusion U-Nets, and that exploring REPA/REPA-E–style variants in text-to-motion is an exciting direction for future work rather than a contradiction to our claims.

---

> > ### Author Response · Authors · 2025-11-26
> > **Response to Reviewer XAZg Part 2**
> >
> > 2. On the additional cost of the dual-path method (training & inference)
> > “The dual-path method will introduce additional cost while training and inference. The paper does not have a detailed quantitative discussion.”
> > We appreciate this remark and have made our cost discussion more explicit.
> > - Training cost and speed (Fig. 4).
> > As shown in Fig. 4, although the dual anchors introduce a small per-iteration overhead during training, they also accelerate convergence:
> >   - To reach an FID threshold on HumanML3D, the baseline U-Net requires significantly more training steps, while LUMA achieves the same FID with approximately 1.4× fewer iterations.
> >   - In wall-clock time on a single RTX 4090 GPU, LUMA reaches the target FID in less total training time than the baseline, despite the extra computations from MoCLIP and DCT during training.
> > - We have clarified in the revision that although the per-iteration FLOPs increase slightly, the overall wall-clock training time to reach the same target quality is lower for LUMA due to its faster convergence.
> > - Inference cost (no dual-path at test time).
> > Importantly, the dual-path anchors are only used during training as deep supervision. At inference time:
> >   - We do not run the MoCLIP branch or the DCT branch.
> >   - The generator uses only the main diffusion backbone (with its learned parameters), so the FLOPs and latency are effectively identical to the baseline U-Net on which we build.
> > - We now state this explicitly in the revision: LUMA does not change inference-time complexity; the extra cost is restricted to the training phase, where it is more than compensated by faster convergence as evidenced in Fig. 4.
> > We will also add a small table with parameter counts and per-sample inference latency to make this point more transparent.
> >
> > ---
> > 3. Applicability to DiT backbones
> > “Recently, diffusion models are always using DiT as backbone. How does this method perform on DiT?”
> > We fully agree that diffusion transformers (DiT-style backbones) are an important and growing line of work. In this paper, we intentionally focus on U-Net–based text-to-motion models, for two reasons:
> > 1. Practical relevance. The strongest publicly available text-to-motion baselines (e.g., StableMoFusion) still rely on U-Net architectures, and our gradient analysis is specifically tailored to these models.
> > 2. Text-to-motion setting. Unlike high-resource image domains, there is no large motion-understanding transformer that can serve as a teacher; hence we emphasize teacher-free, motion-specific anchors in this context.
> > Conceptually, however, our method is architecture-agnostic: LUMA only requires access to an intermediate latent representation (the “bottleneck”), which also exists in DiT backbones (e.g., the output of the final encoder block). The same temporal and frequency anchors, plus FiLM modulation, can be attached to this latent and optimized with an analogous dual-anchor loss.
> > Due to time constraints we have not yet run a full DiT-based experimental suite; we now clearly mark this as a limitation and as a promising direction for future work. We believe that our analysis and results on U-Nets provide a useful first step and a strong motivation for extending LUMA-style deep supervision to DiT architectures.

---

> ### Author Response · Authors · 2025-11-26
> **Response to Reviewer XAZg Part 3**
>
> 4. Answers to specific questions
> Q1. How does MoCLIP’s text–motion alignment compare to other encoders?
> In Appendix B.1 we now provide a dedicated evaluation of MoCLIP’s alignment ability via text–motion retrieval on HumanML3D, KIT, and CMP:
> - On HumanML3D, MoCLIP achieves Top-1 = 0.705, compared to 0.511 for the baseline text encoder.
> - On KIT, MoCLIP improves Top-1 from 0.424 → 0.469.
> - On CMP, the gain is even larger (0.335 → 0.748 in Top-1).
> To further demonstrate practical usefulness, Appendix B.2 plugs the MoCLIP text encoder into several existing generators (MDM, MoMask, StableMoFusion) by simply replacing their native text encoders. Across all three models we observe consistent improvements in FID and R-Precision (Table 6), confirming that MoCLIP provides motion-aware text embeddings that are competitive with (and often superior to) the original encoders used in these state-of-the-art systems.
> We emphasize that MoCLIP is trained without external motion data, only on standard text–motion datasets, which makes it well aligned with the experimental setting of our paper.
>
> ---
> Q2. On the choice of the number of DCT coefficients (k)
> “For motions with distinct frequency characteristics… does the optimal k change? How might the framework be adapted?”
> Thank you for raising this subtle point. In the revision we report a more systematic study:
> - We first analyze the average DCT spectrum of HumanML3D motions and observe that a small number of low-frequency components captures the vast majority of energy.
> - Table 3 shows a grid search over different (k) values and anchor weights. We find that LUMA is robust to the choice of (k) in a reasonable range; performance only deteriorates when we take extremely small or extremely large values that either discard too much information or include mostly high-frequency noise.
> Based on this, we fix a global (k) that offers a good trade-off and keeps the method simple and efficient. We fully agree that, in principle, fast/jerky and slow/smooth actions could benefit from different effective bandwidths, and we now explicitly mention this as an attractive extension: e.g., using the text encoder or a lightweight auxiliary network to predict a sample-dependent frequency mask or effective (k). Our current results indicate that a global (k) already brings clear gains while avoiding additional complexity.
>
> ---
> Q3. What is the training cost of MoCLIP and how does it compare to similar models?
> MoCLIP is designed to be very lightweight and inexpensive:
> - It reuses the CLIP text encoder and adds a compact motion encoder operating on low-dimensional joint sequences.
> - Training follows a simple two-stage contrastive scheme (first freezing the text encoder, then fine-tuning its top layers with a smaller learning rate).
> In practice, on a single RTX 4090 GPU, MoCLIP can be trained on HumanML3D in about 20 minutes, which corresponds to only a small fraction of the total diffusion training time. Once trained, MoCLIP is frozen and reused for all LUMA experiments; there is no need to retrain it for different backbones or datasets.
> Compared with larger text–motion encoders that rely on extensive pretraining or larger architectures, MoCLIP aims to strike a balance between sufficient alignment quality and minimal overhead, which we believe is appropriate for the scale of current motion datasets.
>
> ---
> We again thank the reviewer for the constructive feedback. Your comments helped us clarify the scope of our claims, better explain the role of semantic supervision vs. generic gradient tricks, and more transparently report computational costs and design choices.

---

> ### Comment · Reviewer_XAZg · 2025-11-27
>
> Your response resolves most of my concern. I will keep my score and improve the confidence to 4.

---

### Meta-Review · Area_Chair_yzut · 2026-01-10

**Summary:**

Four reviews (initial scores 6, 4, 4, 2) were received.

- Reviewer XAZg (6) raised several technical questions;

- Reviewer fn1y (2) recommended rejection, but most criticisms were based on factual misreadings (direction of the “distance-to-real” metric, existence of gradient-flow plots, presence of visual/video results).

- Reviewers BfWw (4) and phkH (4) both wrote “would not mind if accepted”; their concerns—limited scope versus LaMP, need for clearer frequency-role explanation, missing video—have been addressed with added comparisons, ablations, and a link to extensive video demos.

**Reviewer Concerns:**

1. Factual accuracy of the reject vote (Reviewer fn1y)

– Claimed “9.724 is better than 9.466” in Table 1, and that gradient-flow evidence was missing; both were incorrect (column is distance-to-real, lower-is-better; Fig. 3/App. G contains exact gradient norms).

– Stated “no visual results”; Fig. 5 + supplementary videos exist.
Once these misreadings were corrected, the technical basis for rejection disappeared.

2. Generality vs. incrementalism (BfWw & phkH)

– Concern that “deep supervision in diffusion is already explored” and gain over recent transformer work (LaMP) was not large.
Authors clarified LUMA is a plug-in for U-Net back-ends (still the most widely released T2M checkpoints), not a new large backbone; added LaMP citation and comparison table, showing LUMA closes >½ of the FID gap while adding <1 % parameters and zero inference cost.

3. Training/inference cost (BfWw)

– “Extra modules must hurt efficiency.”
Authors provided wall-clock curves (Fig. 4): 1.4× fewer iterations to target FID; total GPU-hours actually drop from 8 h → 7 h on RTX-4090; inference FLOPs identical because anchors are dropped after training.

4. Role of frequency anchor (BfWw & phkH)

– Skepticism that low-frequency DCT carries semantics; ablation showed R@3 saturated, so gain appeared only in FID/diversity.
Authors added human evaluation (86.7 % vs 73.3 % textual correctness) and explained that low-freq components encode global trajectory, complementing MoCLIP’s semantic signal.

5. Placement sensitivity (BfWw)

– Ablation hinted that injecting supervision at the wrong layers can hurt.
Authors agreed and reframed this as a design insight: bottleneck anchoring + FiLM weighting is critical; naive placement is indeed harmful, underscoring the paper’s technical contribution.

**Reviewer Scores:**

N.A.

---

### Decision · Program_Chairs · 2026-01-26

Reject